# Distinct stabilization of the human T cell leukemia virus type 1 immature Gag lattice

Martin Obr[1,5], Mathias Percipalle[1], Darya Chernikova [1], Huixin Yang [2], Andreas Thader[1], Gergely Pinke[1], Dario Porley[1], Louis M. Mansky[2], Robert A. Dick[3,4] & Florian K. M. Schur [1] ✉

Human T cell leukemia virus type 1 (HTLV-1) immature particles differ in morphology from other retroviruses, suggesting a distinct way of assembly. Here we report the results of cryo-electron tomography studies of HTLV-1 virus-like particles assembled in vitro, as well as derived from cells. This work shows that HTLV-1 uses a distinct mechanism of Gag–Gag interactions to form the immature viral lattice. Analysis of high-resolution structural information from immature capsid (CA) tubular arrays reveals that the primary stabilizing component in HTLV-1 is the N-terminal domain of CA. Mutagenesis analysis supports this observation. This distinguishes HTLV-1 from other retroviruses, in which the stabilization is provided primarily by the C-terminal domain of CA. These results provide structural details of the quaternary arrangement of Gag for an immature deltaretrovirus and this helps explain why HTLV-1 particles are morphologically distinct.

The Retroviridae family includes two important human pathogens infecting T cells, human immunodeficiency virus type 1 (HIV-1) and human T cell leukemia virus type 1 (HTLV-1). The global prevalence suggests that the number of people living with HTLV-1 (a member of the *Deltaretrovirus* genus) ranges from 5 to 10 million, which is likely an underestimate[1]. While most HTLV-1 infections remain asymptomatic, approximately 5% lead to aggressive diseases such as HTLV-1-associated myelopathy and tropical spastic paraparesis (HAM-TSP) or adult T cell leukemia and lymphoma (ATLL). ATLL is an aggressive form of cancer with a median survival rate of less than 1 year[2,3].

The retroviral structural protein Gag forms the immature protein shell of nascent virus particles[4]. All retroviral Gag proteins contain three canonical domains (Extended Data Fig. 1a): matrix (MA), capsid (CA), consisting of independently folded N-terminal (CA-NTD) and C-terminal (CA-CTD) domains, and nucleocapsid (NC). During immature virus particle formation, these domains function in membrane binding, viral lattice self-assembly and viral genomic RNA (vgRNA) packaging, respectively. Gag oligomerization is primarily driven by interactions between CA domains and these interactions determine virus particle morphology and size[5]. The immature Gag lattice has local six-fold symmetry and is incomplete; for example, in HIV-1, it covers ~60% of the available membrane surface area inside a virion. Upon maturation, the viral-encoded protease cleaves Gag at defined positions, leading to a cascade of structural changes that rearrange the virion interior[6]. In a mature virion, MA remains associated with the viral membrane, while the CA protein forms a core consisting of CA hexamers and pentamers. The CA core contains the condensed NC–vgRNA complex, reverse transcriptase and integrase. Expression of Gag in mammalian cells, as well as the bacterial expression and purification of certain truncated variants thereof, is sufficient for assembly of virus-like particles (VLPs) in vitro[7] that have authentic immature Gag or mature CA architectures[8–11].

Despite substantial sequence variation among retrovirus species, CA shows a strongly conserved tertiary fold, with six to seven α-helices in the CA-NTD and four α-helices in the CA-CTD[12–18]. The latter harbors the highly conserved major homology region (MHR), which has been implicated in preserving the CA-CTD protein fold and in establishing relevant interactions in the assembly of retroviral lattices[9,19]. HIV-1 (*Lentivirus*) and Rous sarcoma virus (RSV; *Alpharetrovirus*) have a Gag cleavage product between CA and NC named the spacer peptide (SP)

[1]Institute of Science and Technology Austria (ISTA), Klosterneuburg, Austria. [2]Institute for Molecular Virology, University of Minnesota, Minneapolis, MN, USA. [3]Department of Molecular Biology and Genetics, Cornell University, Ithaca, NY, USA. [4]Department of Pediatrics, Laboratory of Biochemical Pharmacology, Center for ViroScience and Cure, Emory University School of Medicine, Atlanta, GA, USA. [5]Present address: Material and Structural Analysis Division, Thermo Fisher Scientific, Achtseweg Noord, Eindhoven, Netherlands. ✉e-mail: florian.schur@ist.ac.at

and Mason-Pfizer monkey virus (M-PMV; *Betaretrovirus*) has a similar segment in CA named the spacer peptide-like region[20,21]. The SP region of Gag is important for immature assembly and in the regulation of maturation[4]. Similarly, Gag and CA lattice formation in immature and mature virus particles, respectively, requires CA-CTD dimerization, which is established by hydrophobic residues in helices 8 or 9 of the CA-CTD[22].

The CA-CTD has been proposed to have a dominant role for immature assembly of retroviruses. Variants of HIV-1 Gag containing only the C-terminal half of CA and downstream regions can form VLPs when expressed in cells[23,24]. Cryo-electron tomography (cryo-ET) and subtomogram averaging of purified immature virus particles or in vitro assembled immature-like VLPs of lentiviruses, alpharetroviruses, betaretroviruses and gammaretroviruses demonstrate that CA-CTD dimerization is a key factor in the conserved structural arrangement of the CA-CTD hexamer of immature Gag lattices[8,18,25,26]. These studies also show that the immature CA-NTD arrangement differs substantially among retrovirus genera. In summary, these structural studies are consistent with the hypothesis that the CA-CTD is responsible for forming the essential stabilizing immature assembly interfaces in retroviruses, while the CA-NTD may have a primary role in determining immature lattice curvature and particle size and represents a major binding site for host cell factors[27].

It is not certain whether the function of the CA-NTD in assembly is conserved for retroviruses and whether there are other CA-NTD arrangements that have not been reported. Interestingly, previous mutagenesis experiments targeting residues in the CA-CTD of HTLV-1 Gag did not affect VLP budding[28], suggesting that, unlike other retroviruses, the CA-CTD is not the key determinant of immature assembly. Morphological studies of HTLV-1 particles using cryo-electron microscopy (cryo-EM) also showed peculiar differences in Gag lattice shape, such as areas of flat lattices and varying membrane–CA distance[29–31]. Taken together these studies suggest that HTLV-1 may use different Gag assembly mechanisms than other retroviruses.

However, understanding Gag assembly mechanisms in HTLV-1 is limited by the current lack of structural information for this virus. Given that HTLV-1 is a human pathogen of importance and the ongoing development of inhibitors targeting HIV-1 CA (CA assembly inhibitors)[32], it is critical to generate this structural information.

Here, we report cryo-ET and subtomogram averaging results of full-length and truncated HTLV-1 Gag assemblies. Together with supporting mutagenesis experiments, this allows us to describe the key immature HTLV-1 assembly determinants, delineating the contributions of the CA-NTD and CA-CTD to assembly. Our results show that HTLV-1 uses a distinct CA-NTD arrangement to form the immature Gag lattice and that, in contrast to other retroviral structures, the CA-NTD is the key determinant of immature HTLV-1 assembly.

## Results

### HTLV-1 Gag VLPs display unusual variability

To gain insight into the immature architecture of HTLV-1, we used cryo-ET to visualize mammalian cell-derived Gag-based VLPs (Table 1, Fig. 1a and Extended Data Fig. 1c). As previously reported[30,31], the shape and size of the VLPs varied substantially. The majority of VLPs were spherical; however, the local curvature ranged from flat patches to sharp kinks (Fig. 1a and Supplementary Video 1). As with other retroviruses, the individual domains of Gag formed distinct layers of density starting at the inner leaflet of the viral membrane (corresponding to MA) and extending toward the center of the VLP by ~25 nm (Fig. 1b,c). The CA layer exhibited a regular organization; hence, we used subtomogram averaging (Extended Data Fig. 2a) to obtain higher-resolution information for this layer. Alignments focused on the CA layer resulted in a blurred density for the membrane, which is consistent with the measured variation in the CA-to-membrane distance (mean distance of 17.7 nm, s.d. of ±0.8 nm) (Extended Data Fig. 2b,c). The distance of

**Table 1 | Data acquisition and processing statistics for cryo-ET**

| | HTLV-1 Gag VLPs, (EMD-17941 to EMD-17943) | HTLV-1 MA$_{126}$CANC tubes, (EMD-17929 to EMD-17940) | |
|---|---|---|---|
| **Data collection and processing** | | | |
| System | TFS Titan Krios 3Gi | TFS Titan Krios 3Gi | FEI Titan Krios |
| | System 1 | System 1 | System 2 |
| Detector | Gatan BioQuantum K3 | Gatan BioQuantum K3 | Gatan K2xp |
| Magnification | ×80,000 | ×80,000 | ×80,000 |
| Voltage (kV) | 300 | 300 | 300 |
| Electron exposure (e⁻ per Å²) | ~145 | ~145 | ~145 |
| Dose rate (eps) | 18 | 21 | 5.4 |
| Defocus range (µm) | 1.25–3.5 | 1–3.5 | 1–3 |
| Slit width (eV) | 20 | 20 | 20 |
| Pixel size (Å) | 1.381 | 1.381 | 1.327 |
| Acquisition scheme, tilt | −60/60°, 3 | −60/60°, 3 | −60/60°, 3 |
| Frame number | 8 | 8 | 10 |
| Symmetry imposed | C2 | C2 | C2 |
| Tomograms | 85 | 24 | 45 |
| Final subtomograms (no.) | 264,000 (CA-NTD) / 58,000 (CA-CTD) | 268,000 | |
| Map resolution (Å) | 5.9 (CA-NTD) / 6.2 (CA-CTD) | 3.4 | |
| FSC threshold | 0.143 | 0.143 | |
| **Refinement** | HTLV-1 MA$_{126}$CANC, residues 13–125 (PDB 8PU6) | | |
| Model resolution (Å) | 4/3.4 | | |
| FSC threshold | 0.5/0.143 | | |
| Map sharpening B factor (Å²) | – | | |
| Model composition | | | |
| Nonhydrogen atoms | 2,679 | | |
| Protein residues | 339 (in three chains) | | |
| R.m.s.d. | | | |
| Bond lengths (Å) | 0.005 | | |
| Bond angles (°) | 1.011 | | |
| **Validation** | | | |
| MolProbity score | 1.75 | | |
| Clashscore | 10 | | |
| Poor rotamers (%) | 15 (5%) | | |
| Ramachandran plot | | | |
| Favored (%) | 321 (96%) | | |
| Allowed (%) | 12 (4%) | | |
| Disallowed (%) | 0 (0.0%) | | |

FSC, Fourier shell correlation.

flat Gag lattice regions to the membrane was previously reported to be greater than the distance of curved regions[30].

Retroviral Gag lattices are spherical and the CA layer follows local *C6* symmetry[8,18,33]. However, to account for the observed heterogeneity

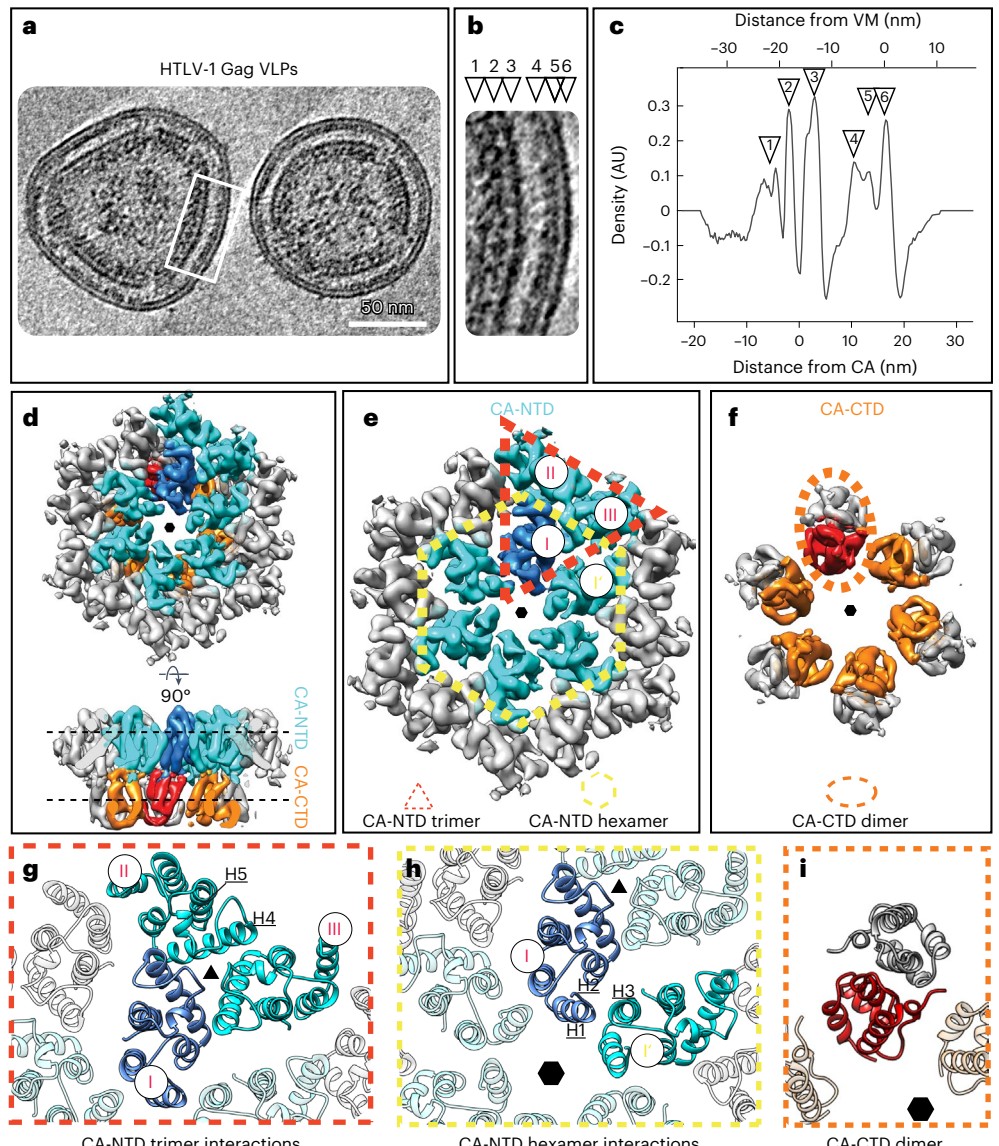

**Fig. 1 | Cryo-ET of immature HTLV-1 Gag VLPs. a**, Computational slice (thickness, 8.8 nm) through a cryo-electron tomogram containing HTLV-1 Gag-based VLPs. Protein density is black. Scale bar, 50 nm. The shown tomogram is representative of the 85 tomograms acquired. **b**, Enlarged view of the Gag lattice within VLPs, as annotated by a white rectangle in **a**. The arrowheads designate the different layers of the radially aligned Gag lattice underneath the viral membrane (VM): NC-RNP (1), CA-CTD (2), CA-NTD (3), MA (4), inner leaflet (5) and outer leaflet (6). **c**, The 1D radial density plot of the Gag lattice in immature HTLV-1 Gag-based VLPs. Two zero-value reference points are reported for the distance measurement. The primary *x* axis reference zero value is set at the VM (top *x* axis). The secondary *x* axis reference zero value is set at the local density minimum between the CA-NTD and CA-CTD (bottom *x* axis) and is provided to allow a straightforward comparison to Fig. 2c. The distance of the different layers from these reference points is given in nanometers. The annotation with arrowheads is as in **b**. AU, arbitrary units. **d**–**f**, Isosurface representations of the subtomogram average of the CA hexamer from HTLV-1 Gag-based VLPs.

The CA-NTDs of the central hexamer are colored cyan, with one monomer highlighted in blue. Two additional CA-NTD monomers from adjacent hexamers are also colored in cyan, to highlight the trimeric interhexamer interface. The CA-CTDs of the central hexamer are colored orange and red. The hexameric arrangement is indicated by a small hexagon. **d**, CA lattice as seen from the outside of the VLP and rotated by 90° to show a side view. **e**, Top view of the CA-NTD, with the trimeric interhexamer interface and the intrahexameric interface indicated with a dashed red triangle and dashed yellow hexagon, respectively. CA monomers in the trimeric interhexamer interface are annotated with I, II and III. **f**, Top view of the CA-CTD hexamer. A dashed orange ellipsoid highlights one CA-CTD dimer linking adjacent hexamers. **g**,**h**, Molecular models of the CA-NTD and CA-CTD rigid-body fitted into the EM density of the immature CA lattice. Coloring as in **d**–**f**. **g**, Trimeric CA-NTD interactions linking hexamers, involving residues spanning helices 4 and 5. **h**, Interactions around the hexamer, involving helices 1, 2 and 3 from adjacent CA-NTDs. **i**, Model of the CA-CTD dimer.

of HTLV-1 Gag VLP shapes, we applied local *C2* symmetry during subtomogram averaging. The resulting maps had a CA-CTD layer that was resolved to a lower resolution than the CA-NTD layer, which we predict is because of heterogeneity caused by flexibility of the CA-CTD. Three-dimensional (3D) classification focused on the CA-CTD yielded a class in which all helices were resolved. In contrast, the CA-NTD did not show increased heterogeneity. Therefore, we used both domains as

independent species in a multiparticle refinement using the software M[34]. This approach yielded 5.9-Å and 6.2-Å resolution maps of the CA-NTD and CA-CTD layers, respectively (Fig. 1d–f and Extended Data Fig. 2d,e). This allowed for the generation of a model of the immature HTLV-1 CA lattice by rigid-body fitting models of HTLV-1 CA-NTD and CA-CTD into the EM density map (Extended Data Fig. 2f). Densities for MA and NC above and below CA, respectively, are present in the

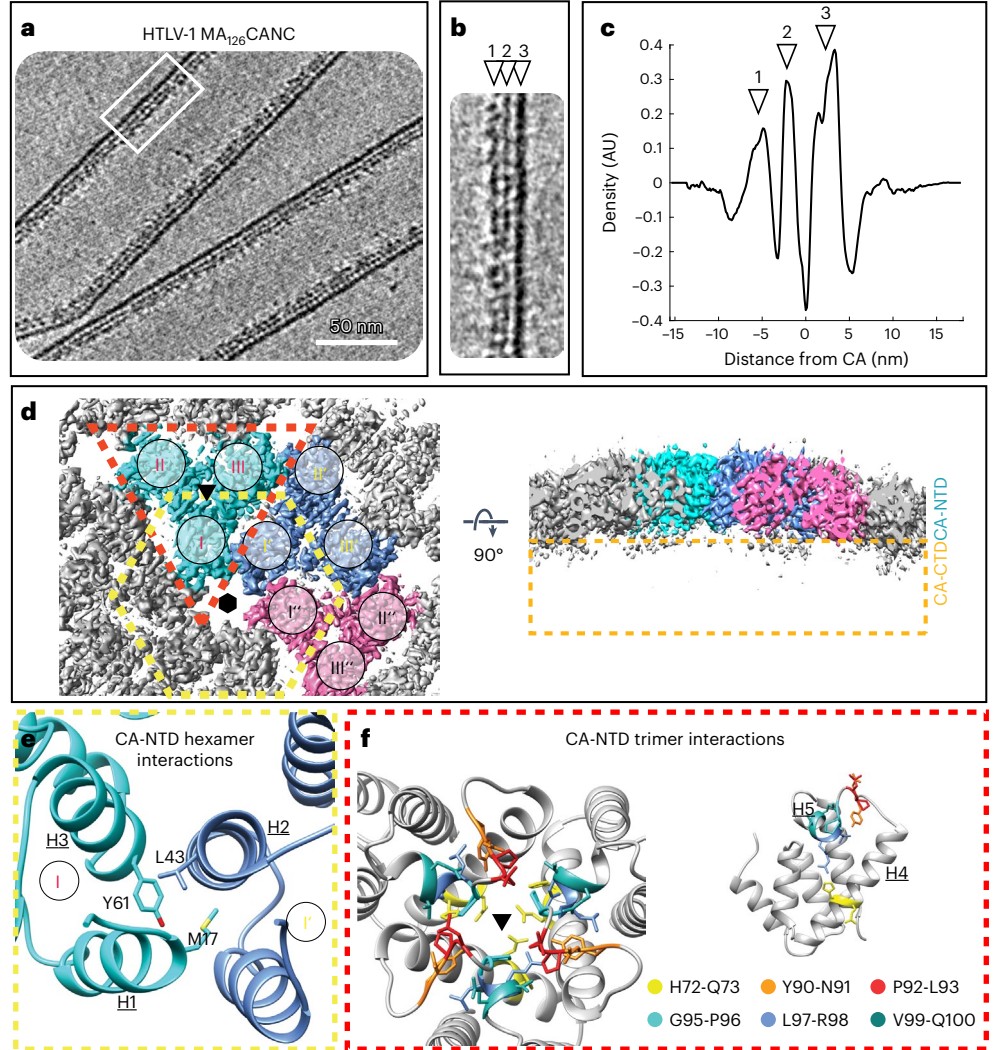

**Fig. 2 | Structural model of the immature HTLV-1 CA-NTD interactions.**
**a**, Computational slice (thickness, 8.8 nm) through a cryo-electron tomogram containing HTLV-1 MA$_{126}$CANC tubes. Protein density is black. Scale bar, 50 nm. The tomogram is representative of the 69 tomograms acquired. **b**, Enlarged view of the CANC lattice within tubes, as annotated by a white rectangle in **a**. The arrowheads designate the different layers of the tubular lattice: NC-RNP (1), CA-CTD (2) and CA-NTD (3). **c**, The 1D radial density plot of the CANC lattice in MA$_{126}$CANC tubes, measuring the distance of the individual CA domains and NC-RNP from the linker between the CA-NTD and CA-CTD. **d**, EM density map of the immature HTLV-1 CA-NTD hexamer at 3.4-Å resolution, seen from the outside of the tube (left) and a side view (right). The three symmetry-independent trimer positions are colored in cyan, blue and pink. The remaining three CA-NTD

domains of the central hexamer are colored in dark gray. Note the missing density for the CA-CTD in the side view on the right. **e**, Zoomed-in view into the CA-NTD intrahexamer interface (annotated with a yellow dashed hexagon in **d**). Labeling is as in Fig. 1. Assembly-relevant residues and the corresponding helices are annotated. **f**, Left: zoomed-in view into the trimeric CA-NTD interface (annotated with a red triangle in **d**). Residues within the trimeric interface that were analyzed using mutagenesis experiments are shown and colored according to the indicated coloring scheme. Right: for simplicity, one CA-NTD domain with the highlighted and colored residues is shown in side view to allow an easier appreciation of the residue location within the CA-NTD. The C2 hexamer center of the isosurface view in **d** is annotated by a schematic hexamer and the trimeric interfaces in **d**,**f** are annotated by black triangles.

one-dimensional (1D) density plot (Fig. 1c) but MA and NC are not resolved in the subtomogram averages, suggesting that neither domain follows the same organization as the CA lattice.

**Two CA-NTD interfaces stabilize the immature lattice**
The CA-NTD establishes lateral interactions that form two basic building blocks shaping the lattice: a CA-NTD trimer and a CA-NTD hexamer (Fig. 1e,g,h and Supplementary Video 2). For both the trimer and the hexamer, each CA-NTD monomer contacts two adjacent monomers around the local pseudo-symmetry axes. The trimer is formed by interactions spanning residues in helices 4 and 5 (Fig. 1e,g). Similarly, the hexamerization interface involves helices 1, 2 and 3 of two neighboring monomers (annotated I and I' in Fig. 1e,h). Unlike in some other retroviruses[8,18,25], there is no dimerization interface in the HTLV-1 CA-NTD.

**CA-CTD forms isolated dimers but not a continuous lattice**
Within the CA-CTD layer, the only intermolecular contact is a dimerization interface formed by helix 8 (Fig. 1f,i and Supplementary Video 2). This is similar to other immature retroviral Gag assemblies, where contacts between hexamers are established by a dimeric interface involving two neighboring CA-CTD monomers (Extended Data Fig. 3a). Unlike other retroviruses, where the CA-CTD forms interhexameric and intrahexameric contacts, the CA-CTD of HTLV-1 only forms interhexameric contacts. The positioning of the individual HTLV-1 CA-CTD dimers places them out of reach of the adjacent dimers around the hexameric CA ring (Extended Data Fig. 3b), not enabling them to contribute to intrahexamerization interfaces forming a lattice.

HTLV-1 lacks a six-helix bundle (6HB), which is reported to be critical for immature CA hexamer formation for other retroviruses.

For HIV-1, the 6HB is formed by residues in CA and SP1 (refs. [9,35]). Similarly, in RSV, M-PMV and murine leukemia virus (MLV) the immature lattices are stabilized by interactions downstream of the CA layer[8,18,25] (Extended Data Fig. 3b). HTLV-1 Gag contains only the canonical Gag domains and is not reported to have a spacer peptide between CA and NC. This is consistent with our observation that there are no ordered densities C-terminal to CA, which would be indicative of the presence of a 6HB or another stabilization-providing region. Hence, the absence of an assembly element such as a 6HB may explain why HTLV-1 has a less well-organized CA-CTD layer.

The HTLV-1 CA-CTD effectively acts to help stabilize the CA-NTD interhexameric interactions. The positions of individual dimers are constrained only by the flexible linker between CA-NTD and CA-CTD. The role of CA-CTD seems to be in further crosslinking the NTD layer, while not constraining the shape of the lattice to a single curvature. Instead, the variable distance between the CA-CTD dimers allows for heterogeneity in particle size and shape.

### High-resolution model of the immature HTLV-1 CA-NTD lattice

To analyze the structure of immature HTLV-1 lattice interfaces in more detail, we engineered an HTLV-1 Gag truncation construct for bacterial expression, purification, in vitro assembly and analysis by cryo-ET. Our construct lacked the N-terminal 125 residues of Gag (MA$_{126}$CANC) (Extended Data Fig. 1a and Extended Data Fig. 4a), similar to other Gag truncation variants that have been previously used to study immature HIV-1, equine infectious anemia virus (EIAV) and M-PMV assemblies[9,20,26].

The purified HTLV-1 MA$_{126}$CANC assembled into hollow tubes (Fig. 2a and Supplementary Video 3), similar to immature-like tubes observed for EIAV, M-PMV and HIV-1 (refs. [20,26,36]). Cryo-ET of the HTLV-1 tubular assemblies revealed a density profile of the CANC layer matching the profile observed for HTLV-1 Gag-based VLPs (compare Figs. 1c and 2c). The MA$_{126}$CANC tubes exhibited varying helical parameters. Tubes were sorted into nine groups and each group was processed separately by subtomogram averaging (Extended Data Fig. 5). The two groups with the largest number of tubes had similar geometry and, thus, they were pooled and processed together. The resulting final reconstruction of the CA-NTD layer had an overall estimated resolution of 3.4 Å (Extended Data Fig. 6a,b). The helical pitch and densities for large side chains were clearly visible at this resolution, which enabled us to build a refined model of the immature CA-NTD lattice interactions (Extended Data Fig. 6c and Table 1). The in vitro assembled tubes (Fig. 2d) had the same arrangement of the CA-NTD layer as the full-length HTLV-1 Gag VLPs (Cα root-mean-square deviation (r.m.s.d.) = 1.2 Å), underscoring the biological relevance of our in vitro system. Hence, our refined model allowed identification of the side chains likely to participate in the CA-NTD interfaces that stabilize the immature lattice.

### Residues involved in stabilizing the immature CA-NTD lattice

Previous cell-culture-based mutagenesis studies of CA revealed the critical role of the CA-NTD for particle morphogenesis, while CA-CTD amino acid substitutions did not affect budding. Specifically, CA-NTD residues M17 and Y61 have been identified as essential for immature HTLV-1 Gag lattice assembly and particle formation[37]. In our model, these residues are located in helices 1 and 3, respectively, which form the hexamerization interface together with helix 2 (Fig. 2e). To further characterize this interface, we substituted residue L43, which interacts with M17 and Y61 in our model. The L43A substitution reduced particle release by more than 50% and further morphological characterization of released L43A particles showed aberrant and scarce patches of Gag lattice underneath the viral membrane (Extended Data Fig. 7a,b and Supplementary Table 1).

Double-alanine swap experiments on a set of adjacent residues were performed to characterize the trimeric CA-NTD interface. Residue pairs H72-Q73, Y90-N91, P92-L93, G95-P96, L97-R98 and V99-Q100 were substituted (Fig. 2f) and screened for altered particle production

efficiency. Previously, the G95-P96 pair was already shown in an insertion–substitution approach to reduce particle formation[28], supporting the importance of this region in VLP assembly. Our mutagenesis experiments revealed a substantial role for the H72-Q73, P92-L93 and V99-Q100 residue pairs in immature Gag assembly and particle release (Extended Data Fig. 7c and Supplementary Table 1). The Y90-N91, G95-P96 and L97-R98 residue pair substitutions not only reduced the particle production but also greatly impacted cellular Gag levels, suggesting these substitutions to also have an effect on expression or stability of the Gag protein. Overall, we conclude from these observations that residues within the trimer interface have important roles for both the behavior of the individual Gag molecules and Gag lattice assembly and stability.

### The CA-CTD layer can adapt to variable curvatures

The CA-CTD layer in the tubular arrays was arranged in a similar way as in the VLPs produced from cells (Fig. 1f and Extended Data Fig. 6d), also forming a layer of disconnected dimers. However, in this case, the qualitative difference between CA-NTD and CA-CTD was even more dramatic. The CA-CTD density in different individual tubes did not permit structural analysis other than coarse positioning of the CA-CTD dimer, as the CA-CTD layer was only visible at lower resolution (compare Fig. 2d and Extended Data Fig. 6d). Nevertheless, this allowed us to assess large-scale changes in the CA layer between the different tube geometries. Unlike the CA-NTD layer, the CA-CTD layer was observed to have larger movement of its domains. While the distance between the adjacent symmetry-independent CA-NTD trimers was nearly constant (42.1–42.6 Å), the distance between symmetry-independent CA-CTD dimers was skewed such that it was decreased in the direction of tube curvature (31.8 Å) and increased along the tube axis (35.6 Å) (Extended Data Fig. 6e). Whether there are different conformations of CA-CTD dimers and to what extent these might contribute to the flexibility or fluidity of the layer could not be discerned from our data.

### Biophysical characterization of HTLV-1 CA and its domains

Given the peculiar behavior of the CA-CTD within the immature lattice, we sought to further characterize full-length CA and its two domains separately using biophysical approaches. To this end, HTLV-1 CA (residues 131–344), CA-NTD (residues 131–258) and CA-CTD (residues 259–344) (Extended Data Fig. 4a) were expressed in *Escherichia coli* and purified. The CA, CA-NTD and CA-CTD proteins were subjected to nano differential scanning fluorimetry (nanoDSF), which monitors changes in intrinsic tryptophan fluorescence (ITF) upon protein unfolding as a function of temperature[38] (Extended Data Fig. 8; see also highlighted tryptophan residues within HTLV-1 CA in Extended Data Fig. 1b). We also conducted backreflection and dynamic light scattering (DLS) experiments to measure turbidity and determine cumulant radius as a means of informing about aggregation and assembly properties. We aimed to characterize the properties of CA and its domains in both a soluble form and a potentially assembled state. To this end, we carried out the measurements at two different time points and protein concentrations: (1) a preassembly condition, directly after transferring the protein to assembly buffer at a concentration normalized for tryptophan content, and (2) a postassembly condition, after incubation under assembly conditions. We note that we were able to obtain reproducible assemblies only for HTLV-1 CA (Extended Data Fig. 4b) but not its domains.

These data revealed the overall similar behavior of full-length CA and CA-NTD, while CA-CTD behaved differently. CA-CTD reproducibly showed the highest melting temperature ($T_m$) in both conditions, suggesting that CA-CTD is the most stable of the three protein variants we tested (Extended Data Fig. 8a and Supplementary Table 2). Both full-length CA and CA-NTD showed a higher propensity to either aggregate or potentially form regular structures (Extended Data Fig. 8b,c), while CA-CTD did not show a substantial increase in turbidity or cumulant radius in any of the conditions tested.

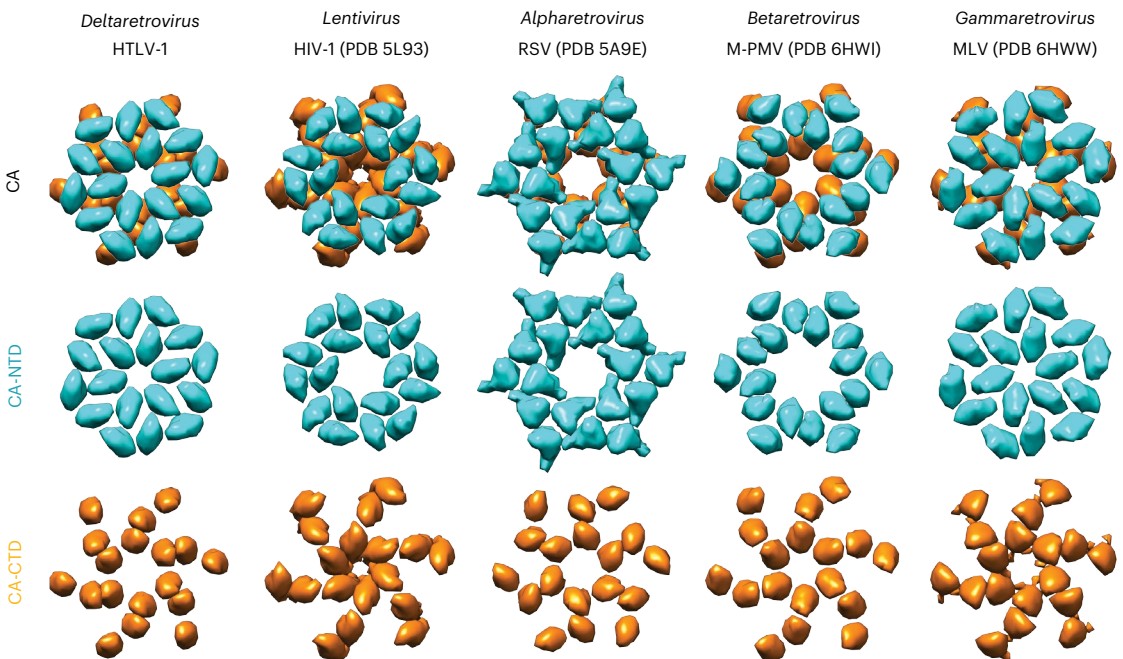

**Fig. 3 | Immature quaternary CA arrangements among retrovirus genera.** Schematic representation of the CA-NTD and CA-CTD arrangements within immature retroviruses from different genera. HTLV-1, *Deltaretrovirus* (as determined in this study); HIV-1, *Lentivirus*; RSV, *Alpharetrovirus*; M-PMV, *Betaretrovirus*; MLV, *Gammaretrovirus*.

For comparison of our HTLV-1 measurements to another retrovirus species, we repeated the same set of experiments with purified HIV-1 CA (residues 133–363), CA-NTD (133–278) and CA-CTD (279–363) (Extended Data Fig. 4c, Extended Data Fig. 8d–f and Supplementary Table 2). Interestingly, HIV-1 CA and its domains behaved almost identically to their HTLV-1 counterparts in the different tested conditions, showing similar trends for melting curves, turbidity and cumulant radius. The minor differences observed were in slight changes in the respective inflection point temperatures and the signal strength for fluorescence ratio, most likely caused by the number of tryptophan residues and their location within the individual CA constructs.

For the tested conditions, we were only able to reproducibly obtain assemblies that could be visualized by negative staining TEM for HIV-1 CA but not the CA-NTD or CA-CTD alone (Extended Data Fig. 4d).

## Discussion

### Noncanonical stabilization of immature HTLV-1 by the CA-NTD

The work presented here expands our knowledge of the immature virion Gag architecture to five of the six genera within the subfamily Orthoretrovirinae (Fig. 3). Importantly, we identified an unconventional mechanism of immature Gag lattice stabilization that is, to date, unique to HTLV-1. Our structural experiments are consistent with a mode in which the CA-NTD is the only lattice-forming CA domain in immature HTLV-1 particles. This is supported by a previous study that did not find substitutions in the CA-CTD affecting particle budding[28] and our study showing that, in our in vitro MA$_{126}$CANC assemblies, the CA-CTD is less organized than the CA-NTD (or the CA-CTD of other retroviruses).

The structural interactions that differentiate retroviral genera can be defined as contributions of the individual interfaces and the orientation of the CA-NTD with respect to the local symmetry axes. The CA-NTD arrangement in HTLV-1 (*Deltaretrovirus*) is similar to that of MLV (*Gammaretrovirus*) (Fig. 3), with both having a tight packing of the trimerization interface and helix 1 pointing toward the center of the hexamer. This results in positioning of CA-NTD in HTLV-1 and MLV closer to that observed in mature CA lattice conformations[18]. When

using *gag* and *pol* sequences for establishing evolutionary relationships, *Lentivirus* and *Betaretrovirus* genera are closer to *Deltaretrovirus* than *Gammaretrovirus*[39]. We performed a multiple-sequence alignment (MSA) for different retroviral CA sequences and generated a phylogenetic tree to evaluate how these evolutionary relationships change when we take into account only CA[40,41] (Extended Data Fig. 9). Indeed, we found the CA relationship for gammaretroviruses and deltaretroviruses to be slightly closer to each other than to lentiviruses, which is the other way compared to full-length Gag comparisons. We interpret this CA phylogeny analysis to be in line with our observation that the quaternary immature CA arrangement in HTLV-1 and MLV is more alike.

### CA domain properties do not solely define assembly behavior

Our biophysical experiments of HTLV-1 and HIV-1 CA and their respective domains indicate that the different assembly properties of HTLV-1 and HIV-1 are not derived solely from features within the individual CA-NTD domains but must also be influenced by a defined interplay between the CA domains. In addition, it is likely that their context within Gag and the interaction with neighboring domains, such as NC and MA, have an important role. NanoDSF does not necessarily consider immature or mature CA behavior but provides insight into protein stability and its propensity to interact or aggregate. Hence, we conclude that the biophysics experiments conducted here provide interesting comparative insights into HTLV-1 and HIV-1 CA but do not necessarily allow further conclusions on how the proteins themselves define immature or mature assembly characteristics.

Future experiments will be required to understand the differential contribution of CA domains to immature assembly in retroviruses. One potential avenue could be to determine high-resolution structures of HTLV-1 and HIV-1 CA chimeras[42] and to solve the structure of the MA lattice in HTLV-1, as recently achieved for HIV-1 (ref. 43).

### Roles of HTLV-1 CA domains in immature lattice stabilization

A unique feature reported for HTLV-1 is the varying distance of the CA layer to the viral membrane in immature Gag-based VLPs. These differences in spacing correspond to the differences in curvature and

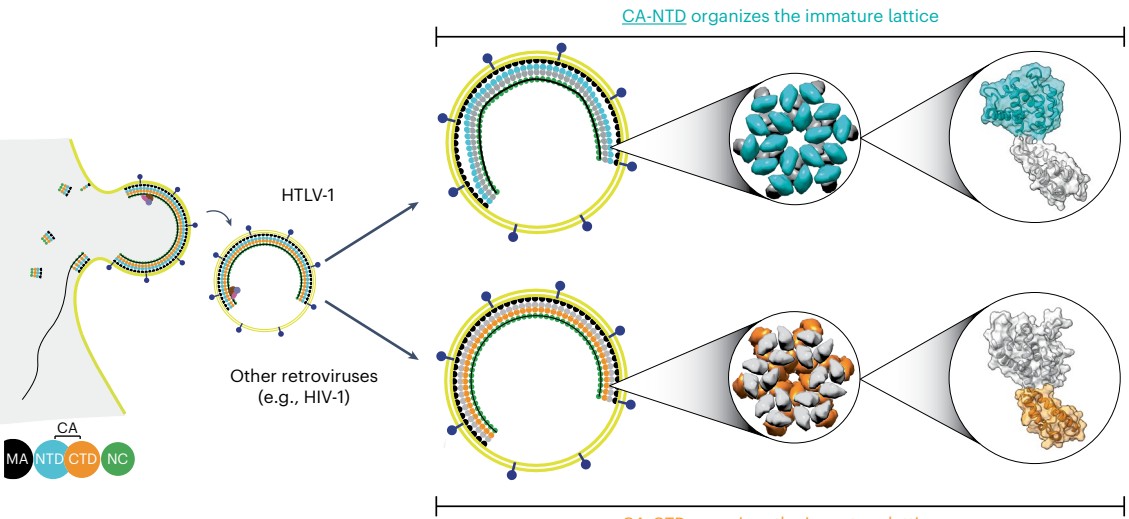

**Fig. 4 | Unconventional immature HTLV-1 CA lattice stabilization.** Schematic representation of the differential role of the CA-NTD and CA-CTD in the assembly of immature HTLV-1 particles, compared to other retroviruses with known immature structures (using HIV-1 as an example). In the case of HTLV-1, immature lattice interactions are driven by the CA-NTD, while the CA-CTD drives particle formation in HIV-1.

flat patches of the HTLV-1 Gag lattice as reported here (Fig. 1a) and previously[30]. Unlike RSV and M-PMV, which also display a larger but uniform distance of their CA layer to the viral membrane[8,44], HTLV-1 does not contain any noncanonical Gag domains between MA and CA. Therefore, the distance between membrane and CA layer is likely determined by specific properties of MA and CA and not the presence of additional domains.

The variable distance of CA to the membrane could be explained by either MA detachment or a substantial structural change of MA. Our data do not provide any evidence for MA detachment from the membrane in areas of increased flexibility, as we regularly can see protein density attached to the inner leaflet of Gag-associated viral membrane (Supplementary Video 1). In addition, previous studies reported HTLV-1 MA to bind to membranes with higher affinity and efficiency but with lower specificity[45–47], arguing against MA detachment from the viral membrane.

A study of HIV-1 and HTLV-1 Gag chimeras showed that the HTLV-1 CA-NTD is the curvature-defining element, while MA regulates the CA–membrane distance[42]. This was convincingly demonstrated, as any construct with the CA-NTD from HTLV-1 contained a flat lattice. A curved membrane over a flat lattice was observed only in those constructs where both CA-NTD and MA from HTLV-1 were present. The clear preference of HTLV-1 CA-NTD to shape lattices into flat regions appears to be compensated for by more flexibility in HTLV-1 Gag layers both above and below it. On the proximal side, MA seems to tolerate a larger membrane–CA distance range (Extended Data Fig. 2b), allowing to speculate whether MA can undergo a conformational change. On the distal side, CA-CTD forms sparser lateral interactions than other retroviruses, which makes the location of CA-CTD dimers less constrained with respect to the CA-NTD network. This is apparent from the blurring of the CA-CTD layer in our subtomogram averages. Accordingly, residues in the MHR, which have been implicated in immature assembly for most retroviruses, are not positioned to form interactions stabilizing the immature HTLV-1 CA-CTD hexamer. To date most structurally studied retroviruses have a stabilization element, which forms at the six-fold symmetry axis below the CA-CTD[9,18,25,35,48,49]. Our results here find no such domain in HTLV-1, further highlighting that HTLV-1 is structurally unique.

The fundamental difference in the organization of the CA-CTD in HTLV-1 compared to other retroviruses is quite striking. While the CA-CTD is a critical lattice-forming element for other retroviruses[6,23,36,50], some of the canonical CA-CTD functions are taken up by the CA-NTD in HTLV-1 (Fig. 4).

### The role of the HTLV-1 CA-CTD in immature HTLV-1 assembly
This raises a question regarding the function of the CA-CTD in immature HTLV-1 assembly. One possibility is that, in HTLV-1, a direct interaction between the NC–RNA complex and the CA-CTD may contribute to immature lattice formation. In support of this hypothesis, the HTLV-1 CA-CTD possesses a stretch of positively charged residues at its base (toward the VLP center) in helices 9 and 10 that could interact with NC and/or RNA (see underscored residues in Extended Data Fig. 1b). This would be similar to what was previously shown for M-PMV, where a basic RKK motif at the CA-CTD base is involved in promoting virus assembly and RNA packaging[51]. EIAV also has a basic motif, RHR, and density is observed to interact with this motif in slightly acidic in vitro assembly conditions[26], which is consistent with the proposed model.

Moreover, the C-terminal region of the NC domain of HTLV-1 is negatively charged, a feature absent in other retroviruses. Accordingly, it has been shown that HTLV-1 NC is a weak RNA chaperone and nucleic acid binder[52]. The authors posed the hypothesis that the NC C terminus interacts intramolecularly with its zinc fingers effectively blocking nonspecific RNA binding. It was further suggested that, after Gag oligomerization into a lattice, lateral intermolecular NC interactions form between the C terminus and zinc fingers. Additional studies focusing on the HTLV-1 CA-CTD, especially considering the lack of exhaustive CA-CTD mutagenesis experiments, are warranted to more accurately determine the interactions in this region of the immature lattice.

### Conclusions
HTLV-1 infectious spread occurs predominantly by cell-to-cell contact and not by cell-free virus infection of permissive T cells. The implication of the unique features of HTLV-1 Gag lattice stabilization shown here on HTLV-1 cell–cell transmission is unclear. Future work using cellular tomography on assembling and budding virions could provide a clearer understanding of the structural intermediates in the HTLV-1 lifecycle that are important in addressing this outstanding question in the field.

In addition to HTLV-1, the *Deltaretrovirus* genus contains multiple members, such as bovine leukemia virus. Structural comparison of the immature Gag lattice in other deltaretroviruses could help us better understand the general assembly principles and could clarify whether the observed dominance of the HTLV-1 CA-NTD in the immature Gag

lattice is conserved within the genus. Our findings provide a structural basis for guiding future studies on HTLV-1 assembly and maturation, as well as for guiding the discovery of targets for therapeutic intervention. The distinct and unique nature of the HTLV-1 lattice, that is, the absence of hexameric CTD interactions and of a clear inositol hexakisphosphate (IP6)-binding site as observed in HIV-1 (ref. 53), will necessitate alternative drug targeting approaches than, for example, the structure-based maturation inhibitors developed against HIV-1 (refs. 54–56).

## Online content

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

## Methods

### Mammalian cell culture and Gag-based VLP production and purification

The codon-optimized HTLV-1 Gag sequence, subcloned into the pCMV expression vector was ordered from Thermo Fisher Scientific. HEK293T cells (American Type Culture Collection (ATCC), CRL-3216), cultured in 10 × 10-cm dishes, were transfected with the expression plasmid using Lipofectamine LTX Reagent with PLUS Reagent (Thermo Fisher Scientific, 15338100) according to the manufacturer's instructions. Cultivation medium containing released HTLV-1 Gag-based VLPs was obtained 40 h after transfection by ultracentrifugation. This and all further centrifugation steps were performed at 4 °C. Cultivation medium was first clarified by centrifugation for 5 min at 1,500g and subsequent filtering through a syringe-mounted PVDF filter with 0.45-μm pore size (Merck Millipore). VLPs were pelleted through a 20% sucrose cushion at 125,000g for 120 min. The pellet was briefly air-dried and resuspended in 150 μl of PBS. Pooled and resuspended pellets were applied on a 6–18% Optiprep gradient and centrifuged at 235,000g for 90 min. Fractions containing Gag-based VLPs were pooled, diluted 1:4 in PBS and pelleted without a cushion at 260,000g for 45 min. The final pellet was then resuspended in 12 μl of PBS.

### Cryo-ET for HTLV-1 Gag-based VLPs

**Grid preparation for HTLV-1 Gag-based VLPs.** The resuspended Gag-based VLP pellet was kept on ice and mixed with 10-nm colloidal gold suspension before vitrification. Then, 2.5 μl of solution was applied on glow-discharged C-flat 2/2 3C grids and vitrified in liquid ethane using a Leica EM GP2 plunge-freezer. Grids were blotted in back-side mode for 3–4 s. The humidity chamber was conditioned to 8 °C and 90% relative humidity. Grids were stored in liquid nitrogen until imaging.

**Imaging of HTLV-1 Gag-based VLPs.** Data were collected on a Thermo Fisher Scientific Krios G3i equipped with Gatan Quantum K3 using SerialEM for data acquisition[57]. Areas of interest for high-resolution data collection were identified in low-magnification montages. Before tomogram acquisition, gain references were acquired and the filter was fully tuned using DigitalMicrograph. Microscope tuning was performed using the FEI AutoCTF software or SerialEM tuning functions. The slit width of the filter was set to 20 eV. The magnification was set to ×80,000, resulting in a pixel spacing of 1.381 Å. Tilt images were acquired as movies in super-resolution mode consisting of eight dose fractions. The dose rate was set at ~18 electrons per pixel per s. Tilt series were acquired using a dose-symmetric tilt scheme[58], with a tilt range from 0° to −60° and +60° in 3° steps, and at nominal defocus between −1.25 and −3.5 μm. The total dose per tilt image was 3.5 e⁻ per Å². The total dose per tilt image was $3.5\,\text{e}^-$ per $\text{Å}^2$.

**Cryo-ET image processing of HTLV-1 Gag-based VLPs.** A schematic overview of the individual processing steps for the HTLV-1 Gag-based VLPs is shown in Extended Data Fig. 2a. Tilt images with blocked field of view were removed before further steps. Tilt series were aligned using the Etomo package (as part of the IMOD package version 4.9.12) and exposure-filtered, as published previously, using a custom Matlab script[9,59,60]. Contrast transfer function (CTF) estimation was performed using ctffind (4.1.10).

Template matching was performed essentially as previously published[61]. A template was generated using subtomogram alignment of the CA layer from round VLPs from 12 tomograms. A cylindrical mask centered at a CA hexamer and encompassing two adjacent CA rings was applied to the template. In-plane and cone angles were scanned in steps over a range of 180° and 360°, in 6° and 8° steps, respectively. Lattice connectivity analysis (as described previously[61]) was performed to remove peaks outside of lattice and annotate VLPs. The following criteria were used to assess pairs of local cross-correlation maxima: the minimum and maximum spacing between hexamers was 45 Å and

110 Å; the minimum and maximum curvature was −15° and 25°. Only networks containing >20 cross-correlation peaks were considered as patches of lattice and used for subsequent steps.

For initial subtomogram alignment, tomograms were reconstructed in novaCTF[62] using two-dimensional (2D) CTF correction of 2× binned stacks with the multiplication algorithm. Bin8 and bin4 alignments were performed in Dynamo (version 1.1.133)[63] and subTOM[26]. A conservative low-pass filter of ≥25 Å was used during these alignments. Given the irregular and variable shape of Gag-based VLPs, we refrained from using C6 rotational symmetry, which is conventionally used for processing of immature retroviral lattices. Instead, we used C2, which allows for capturing CA hexamer deformation. After bin4 alignments, the VLP list was manually curated to remove particle formations outside enveloped VLPs.

Preprocessing for the Warp-RELION-M pipeline was performed in Warp (version 1.0.9)[64]. Tilt series alignments from Etomo (IMOD version 4.9.12) and subtomogram alignments from Dynamo (version 1.1.133) and subTOM were imported and used for subtomogram reconstruction. The 3D refinement was performed in RELION 3.0.8 at bin4 and bin2, using subtomograms reconstructed in Warp at the respective binning. The 3D classification was performed in RELION 3.0.8 (ref. 65) at bin2 to select a more homogeneous CA-CTD population. A cylindrical mask encompassing six CA-CTD dimers was used during the classification. For both CA-NTD and CA-CTD, a separate species was set up masking the respective layer. The CA-NTD species contained all particles from the 3D refinement at bin2, whereas the CA-CTD species contained just particles from a selected class (Extended Data Fig. 2a). Five rounds of multiparticle refinement were performed, refining particle positions, 2D and 3D warping and CTF parameters. Frame alignment refinement was performed in the last two rounds.

### Model building for the immature HTLV-1 CA lattice from the Gag-based VLP-derived structure.

The model of the immature HTLV-CA assembly was generated by rigid-body fitting the HTLV-1 CA-NTD derived from Protein Data Bank (PDB) 1QRJ (ref. 12) into our EM density using the fit in map option in UCSF Chimera. As we noted that the experimentally derived HTLV-1 CA-CTD from PDB 1QRJ resulted in severe clashes on the CA-CTD dimer interface upon rigid-body fitting, we used a computationally predicted model of the CA-CTD (CA residues 129–207) using ColabFold[66]. Interestingly, this predicted model fitted into our EM density better and resulted in fewer clashes across the dimeric interface. Hence, we used this rigid-body-fitted prediction for modeling the CA-CTD.

### Protein purification and in vitro assembly

**Cloning of truncated Gag constructs for in vitro assembly.** The sequences encoding HTLV-1 MA₁₂₆CANC (Gag residues 126–344), HTLV-1 CA (131–344), HTLV-1 CA-NTD (131–258) and HTLV-1 CA-CTD (259–344) were cloned into pET28 expression vector in frame with 6xHis-SUMO tags using standard molecular cloning methods. The same procedure was done for cloning HIV-1 CA (133–363), HIV-1 CA-NTD (133–278) and HIV-1 CA-CTD (279–363).

**Protein expression and purification of HTLV-1 MA₁₂₆CANC.** An overnight culture of BL21 carrying the pET28/His-SUMO-MA₁₂₆CANC construct was prepared to inoculate a total of 2 L of lysogeny broth (LB) medium supplemented with kanamycin. Following incubation at 37 °C with shaking at 210 r.p.m. until the bacterial culture reached an optical density at 600 nm (OD₆₀₀) of 0.5–0.7, protein expression was induced by the addition of IPTG at a final concentration of 1 mM. Induction was carried out for 6 h at 37 °C with shaking at 210 r.p.m.

Bacterial cells were harvested by centrifugation at 6,000g for 15 min. The resulting cell pellet was dissolved in resuspension buffer (20 mM Tris, 500 mM NaCl, 2 μM ZnCl₂, 5% glycerol, 1 mM PMSF and 1 mM TCEP, pH 8.0).

Protein was extracted from cells by cell lysis through three freeze–thaw cycles involving freezing at −80 °C followed by thawing at 42 °C. After 45–60 min of centrifugation of the lysate at 21,000$g$ and 4 °C, the supernatant was collected and subjected to nucleic acid precipitation by treatment with 10% PEI (polyethylenimine) at a final concentration of 0.3% under stirring at 4 °C for 30 min. Afterward, the mixture was centrifuged for 10 min at 10,000$g$ and 4 °C. The remaining supernatant was treated with ammonium sulfate at a final concentration of 40% under stirring at 4 °C overnight for precipitation of His-SUMO-MA$_{126}$CANC. Precipitated protein was collected by centrifugation at 10,000$g$ and 4 °C for 20 min.

Purification of His-SUMO-MA$_{126}$CANC was carried out using anion-exchange chromatography and affinity chromatography. For anion-exchange chromatography the protein was dissolved in buffer composed of 20 mM Tris and 2 mM TCEP at pH 8.0 and was then applied to a 5-ml HiTrap SP HP column (Cytiva, 17115201) before being eluted with high-salt buffer composed of 20 mM Tris, 500 mM NaCl and 2 mM TCEP at pH 8.0.

The protein sample was then transferred to a 1-ml HisTrap FF column (Cytiva, 17531901) for protein purification by affinity chromatography. Bound protein was treated with wash buffer (20 mM Tris, 500 mM NaCl, 20 mM imidazole and 2 mM TCEP, pH 8.0) before it was eluted with high-concentration imidazole buffer (20 mM Tris, 500 mM NaCl, 250 mM imidazole and 2 mM TCEP, pH 8.0).

To remove imidazole, the protein solution was dialyzed overnight at 4 °C in tubes of the Pur-A-Lyzer Maxi 6000 dialysis kit (Sigma, PURX60100-1KT) against dialysis buffer (20 mM Tris, 500 mM NaCl and 0.5 mM TCEP, pH 8.0). During dialysis, the sample was treated with N-terminally His-tagged Ulp1 protease for removal of the His-SUMO tag from MA$_{126}$CANC. The protease was later removed by reapplication of the sample to a 1-ml HisTrap FF column resulting in high-affinity binding of His-Ulp1 to the nickel Sepharose resin. MA$_{126}$CANC protein exhibits low affinity for the packing material of this column. Hence, it was eluted with 20 mM imidazole wash buffer. After another dialysis step, 150–200-µl aliquots of purified protein were flash-frozen in liquid nitrogen and stored at −80 °C.

**Assembly of MA$_{126}$CANC tubes.** Protein stored in dialysis buffer of 500 mM NaCl was first mixed with an equal volume of salt-free buffer (20 mM Tris and 0.5 mM TCEP, pH 8.0). The protein sample was then transferred into a Pierce concentrator tube with a 10-kDa molecular weight cutoff (MWCO) (Thermo Fisher Scientific, 88513) and protein concentration was increased to 12–20 mg ml$^{-1}$ by centrifugation at 15,000$g$ and 4 °C for 12 min. Then, 10 µl of protein (120–200 µg) was mixed with 2 µl of 5 mg ml$^{-1}$ GT50 nucleotides, 1 µl of 50 mM EDTA and 37 µl of salt-free buffer.

The reaction mixture with a final NaCl concentration of 50 mM was incubated overnight at 4 °C. After incubation, an aliquot of the sample was subjected to negative staining for confirmation of VLP presence by transmission EM on a Tecnai T10.

**Protein expression and purification of HTLV-1 CA, CA-NTD and CA-CTD.** Overnight cultures of BL21 carrying pET28a vectors containing a His-SUMO tag, with HTLV-1 CA, CA-NTD and CA-CTD inserts were grown at 37 °C and 220 r.p.m. until an OD$_{600}$ of 2.5 was reached. These were used to inoculate 1 L of LB medium for each construct and grown at 37 °C and 220 r.p.m. until an OD$_{600}$ of 0.6 was reached, after which protein expression was induced with 1 mM IPTG for 6 h. The cultures were centrifuged at 6,000$g$ and 4 °C for 15 min to collect the cell pellets, which were subsequently dissolved in lysis buffer (20 mM Tris, 500 mM NaCl, 2 µM ZnCl$_2$, 5% glycerol, 1 mM PMSF and 1 mM TCEP, pH 8.0) and subjected to 3–5 freeze–thaw cycles to disrupt the cells.

The resulting lysed cells were centrifuged at 21,000$g$ for 45 min to separate protein from insoluble cell debris, followed by protein purification from nucleic acid by precipitating the nucleic acid with 10% PEI at a final concentration of 0.3% for 10 min under stirring at 4 °C.

The solutions were centrifuged again, at 10,000$g$ and 4 °C for 10 min to collect the precipitated nucleic acid and ammonium sulfate was subsequently added to the supernatant to a saturation of 40% under stirring at 4 °C overnight to precipitate the proteins. The solutions were centrifuged again at 13,000$g$ to collect the proteins in a pellet before resuspending the protein pellets in wash buffer (20 mM Tris, 20 mM imidazole, 500 mM NaCl and 2 mM TCEP, pH 8).

The proteins were then purified using affinity chromatography with a 1-ml HisTrap FF column according to the manufacturers' instructions (Cytiva) and elution buffer containing 20 mM Tris, 250 mM imidazole, 500 mM NaCl and 2 mM TCEP, pH 8.

Following purification, the His-SUMO tag was cleaved from the proteins using Ulp1 protease and the proteins placed for dialysis at 4 °C overnight in dialysis buffer (20 mM Tris, 500 mM NaCl and 5 mM TCEP, pH 8). A second affinity chromatography purification step with Ni-NTA columns was performed to purify the proteins from the cleaved His-SUMO tag, followed again by dialysis in dialysis buffer at 4 °C overnight.

**Protein expression and purification of HIV-1 CA, CA-NTD and CA-CTD.** *E. coli* BL21 cells were transformed with the respective pET28 vectors and grown on agar plates supplemented with 50 µg ml$^{-1}$ kanamycin. Overnight cultures of the BL21 cells carrying the respective vectors were used to inoculate 1 L of LB medium supplemented with kanamycin and 10 mM HEPES. The cultures were grown at 37 °C and 220 r.p.m. until an OD$_{600}$ of 0.9–1.1 was reached, whereby protein expression was induced with 0.3 mM IPTG and protein expressed for 5 h at 30 °C and 200 r.p.m.

The cultures were centrifuged at 6,000$g$ and 4 °C for 15 min to collect the cell pellets that were subsequently dissolved in lysis buffer (20 mM Tris, 500 mM NaCl, 2 µM ZnCl$_2$, 5% glycerol, 1 mM PMSF and 5 mM β-mercaptoethanol, pH 8.0) and subjected to 3–5 freeze–thaw cycles to disrupt the cells.

The resulting lysed cells were centrifuged at 21,000$g$ for 45 min to separate protein from insoluble cell debris, followed by protein purification using affinity chromatography with a 1-ml HisTrap™ FF column according to the manufacturer's instructions (Cytiva) and elution buffer containing 20 mM Tris, 250 mM imidazole, 500 mM NaCl and 2 mM TCEP at pH 8. Following purification, the His-SUMO tag was cleaved from the proteins using Ulp1 protease and the proteins placed for dialysis at 4 °C overnight in dialysis buffer (20 mM Tris, 500 mM NaCl and 5 mM TCEP, pH 8). A second affinity chromatography purification step with Ni-NTA columns was performed to purify the proteins from the cleaved His-SUMO tag. Proteins were then snap-frozen and stored at −80 °C.

**Assembly reactions and preparation for nanoDSF.** HTLV-1 and HIV-1 CA, CA-NTD, and CA-CTD were separately concentrated in Pierce concentrator tubes (10-kDa MWCO for CA and CA-NTD and 3-kDa MWCO for CA-CTD). Concentrations of monomeric HTLV-1 CA, CA-NTD and CA-CTD were normalized to the tryptophan content of the respective constructs, with 0.96 mg ml$^{-1}$ for CA, 1.15 mg ml$^{-1}$ for CA-NTD and 0.70 mg ml$^{-1}$ for CA-CTD under preassembly conditions (50 mM MES and 200 mM NaCl, pH 6) in a volume of 10 µl and placed in Prometheus Series high-sensitivity capillaries (NanoTemper, PR-C006). The same was performed for HIV-1 CA (0.82 mg ml$^{-1}$), CA-NTD (0.66 mg ml$^{-1}$) and CA-CTD (2 mg ml$^{-1}$) but with a different preassembly buffer (50 mM MES, 150 mM NaCl and 5 mM β-mercaptoethanol, pH 6).

For measurement of HTLV-1 postassemblies, the proteins were first placed in assembly buffer (50 mM MES and 200 mM NaCl, pH 6) with a final concentration of ~680 µM and were incubated at 4 °C for 4 h and 26 °C overnight for assembly to occur. A volume of 10 µl was then placed in Prometheus Series high-sensitivity capillaries (NanoTemper, PR-C006). For visualizing assemblies of HTLV-1 CA on a Tecnai T10

transmission EM instrument, an aliquot was used for negative staining with 2% uranyl acetate (UA).

Similarly, for measurement of HIV-1 postassemblies, protein was concentrated to 4 mg ml$^{-1}$ (150 μM) and dialyzed into 50 mM MES, 150 mM NaCl and 5 mM β-mercaptoethanol at pH 6 overnight. IP6 was then added to a final concentration of 4 mM and the protein was incubated at 30 °C for 2 h. A volume of 10 μl was then placed in Prometheus Series high-sensitivity capillaries (NanoTemper, PR-C006). For visualizing assemblies of HIV-1 CA on a Tecnai T10 transmission EM instrument, an aliquot was used for negative staining with 2% UA.

**NanoDSF, backreflection and static light scattering.** The capillaries were loaded into a Prometheus Panta instrument (NanoTemper Technologies). NanoDSF, backreflection and DLS were measured continuously over a temperature ramp of 5 °C min$^{-1}$ from 15 °C to 95 °C at 40% light-emitting diode excitation power. All scans were single reads of three replicates performed at least three times. Data were analyzed using custom Python scripts (Python version 3.11.4).

### Cryo-ET for HTLV-1 MA$_{126}$CANC tubes

**Grid preparation for MA$_{126}$CANC tubes.** After in vitro assembly, MA$_{126}$CANC tubes were kept at 4 °C until plunge-freezing. Then, 2/2 3C C-Flat grids coated with a 2-nm support carbon layer were glow-discharged in the presence of amylamine. Grids were then incubated on a 5-μl drop of sample for 10 min. The rest of the sample was mixed with 10-nm colloidal gold and 2.5 μl of this solution was added to the incubated grids before vitrification. The samples were vitrified in liquid ethane using a Leica GP2 plunger with front-side blotting (blot time, 3–4 s; humidity, 90–95%; temperature, 10 °C) and stored in liquid nitrogen until imaging.

**Imaging of MA$_{126}$CANC tubes.** For in vitro assembled MA$_{126}$CANC tubes, we acquired data on two systems. One dataset was collected on a Thermo Fisher Scientific Krios G3i equipped with Gatan Quantum K3 (system 1). The second dataset was collected on a FEI Titan Krios, operated at 300 keV, equipped with a Gatan Quantum 967 LS energy filter and a Gatan K2xp direct electron detector (system 2). The slit width of the filter was set to 20 eV on both systems. SerialEM was used for data acquisition in both cases. Areas of interest for high-resolution data collection were identified in low-magnification montages. Before tomogram acquisition, gain references were acquired and the filter was fully tuned. Filter tuning was performed in DigitalMicrograph. Microscope tuning was performed using the FEI AutoCTF software or SerialEM tuning functions.

On system 1, the magnification was set to ×80,000, resulting in a pixel spacing of 1.381 Å. Tilt images were acquired as movies in super-resolution consisting of eight dose fractions. The dose rate was set at -21 eps. On system 2, the magnification was set to ×105,000, resulting in a pixel spacing of 1.327 Å. Tilt images were acquired as movies in super-resolution consisting of ten dose fractions. The dose rate was set at -5.4 eps.

Tilt series were acquired using a dose-symmetric tilt scheme[58], with a tilt range from 0° to −60° and +60° in 3° steps, at a nominal defocus between −1.5 and −3.5 μm. The total dose per tilt image was 3.5 e$^-$ per Å$^2$.

**Cryo-ET image processing of MA$_{126}$CANC tubes.** A schematic overview over the individual processing steps for the MA$_{126}$CANC tubes is shown in Extended Data Fig. 5a. Before tilt series alignment, movies were aligned using the IMOD 'alignframes' function. Tilt images with a blocked field of view were removed before further steps. Tilt series were aligned using the Etomo (part of the IMOD package, version 4.9.12) and exposure-filtered using a custom Matlab (version R2018b) script[9,59]. CTF estimation was performed using ctffind (4.1.10).

For initial subtomogram alignment described below, CTF-corrected tomograms were reconstructed in novaCTF using the

multiplication algorithm with a slab size of 15 nm. Bin8 and bin4 alignments were performed in Dynamo (version 1.1.133) and subTOM. A conservative low-pass filter of ≥25 Å was used during these alignments.

Preprocessing for the Warp-RELION-M pipeline was performed in Warp. Tilt series alignments from Etomo and subtomogram alignments from Dynamo and subTOM were imported and used for subtomogram reconstruction. The 3D refinement was performed in RELION 3.0.8 at bin4 and bin2, using subtomograms reconstructed in Warp at the respective binning.

**Classification of tube geometries for structure determination.** Because the tube geometry was variable for the CANC tubes, tubes were sorted into groups containing similar tube geometries. Specifically, the angle between the tube axis and the hexamer–hexamer connection was used to distinguish between different tube geometries (Extended Data Fig. 5b).

To this end, first, a de novo reference was generated for each tube separately by subtomogram alignment of regularly spaced volumes along a given tube using Dynamo and subTOM. The initial alignments were performed at bin8. The first step was to remove particles that converged to the same spatial position using distance cleaning. A cross-correlation threshold was set manually for each tomogram upon visual inspection to remove misaligned and bad subvolumes. The positions of subtomograms derived from the initial alignments were used for geometry analysis. Tubes were sorted into groups such that the maximum angle difference within one group was 5°. Subtomogram alignment and tube grouping were then refined at bin4.

**Merging datasets for multiparticle refinement.** Because the datasets acquired on the two microscope systems differed in pixel size, they were processed separately until the multiparticle refinement step. Then, because the pixel size of system 1 was not calibrated, we used the fitmap function of UCSF Chimera[67] to maximize the overlap between maps generated using systems 1 and 2, while varying the pixel size of system 1. For further processing, we used this estimated pixel size rather than the nominal value. On this basis, we estimated the actual pixel size on system 1 to be 1.339 Å. To unify the pixel size used for multiparticle refinement, we used a box size that yielded an integer upon dividing with both pixel sizes. A box size of 321.3 Å fulfilled the criteria.

**Multispecies refinement of MA$_{126}$CANC tubes.** The 3D refinement in RELION 3.0.8 was performed for each group of tubes separately at bin4 and bin2. Subtomograms were reconstructed in Warp. The respective halfmaps and particle coordinate star files were used as a starting point for multiparticle refinement in M, gradually adding refinement options. Five rounds of refinement were performed as described previously[68]. Then, 2D and 3D warping was performed in the first round, particle pose refinement was added in the second round, CTF refinement was added in the third round and stage angle refinement and movie refinement were added in the fourth and final iterations. This yielded nine CA-NTD hexamer structures at resolutions from 3.7 to 7.0 Å. To increase the resolution for model building, the two most populated groups sharing similar geometry were pooled, subtomograms were reconstructed using M at a pixel size of 1.77 Å and the pooled population was subjected to Autorefine 3D in RELION. Afterward, two final rounds of M refinement were performed. The final 3D refinement was performed in RELION, with subtomograms reconstructed using M at bin1.

**Model refinement.** The initial model for HTLV-1 CA-NTD was obtained by adjusting PDB 1QRJ, by flexibly fitting it into the EM density using the Isolde plugin of ChimeraX[69,70]. The final model was then built by iterating between real-space refinement in PHENIX[71] and manual adjustments in Coot[72].

## Data visualization and figure preparation

The 3D volumes were visualized using UCSF Chimera (version 1.17.3)[67], ChimeraX (versions 1.7.1 and 1.8)[70] and IMOD (version 4.9.12)[60]. Images of 3D volumes and models were made using Chimera or ChimeraX or by ray tracing in PyMOL (version 1.3r1; Schrödinger). Figures were prepared using Adobe Illustrator. Videos were generated in ChimeraX and Adobe Premiere Pro 2023.

The 1D density plots were calculated from the bin1 (CANC tubes) or bin2 (Gag VLPs) unsharpened unmasked average, assuming a tube diameter of 60 nm and VLP diameter of 130 nm.

## Mutagenesis of residues involved in stabilizing the immature CA-NTD

To characterize the residues involved in stabilizing the immature CA-NTD, we conducted mutagenesis on key residues. First, we substituted residue L43 to alanine, as L43 may contact M17 and Y61. Second, we conducted mutagenesis of the trimeric CA-NTD interface. Specifically, residue pairs H72-Q73, Y90-N91, P92-L93, G95-P96, L97-R98 and V99-Q100 in the CA-NTD were created using the Gibson assembly method[73]. The correct creation of the mutants was verified by Sanger DNA-sequencing analysis. The efficiency of immature particle production was analyzed by quantifying Gag proteins in released particles through harvesting cell culture supernatants and using immunoblot analysis with a mouse monoclonal anti-HTLV p24 (CA) antibody (Santa Cruz Biotechnology, sc-53891). To determine the particle production (that is, particle release) efficiency of mutants relative to that of the wild type (WT), the Gag expression levels detected in cell lysates of the WT and mutants were normalized relative to the respective glyceraldehyde 3-phosphate dehydrogenase (GAPDH) level. The mutant particle release relative to that of the WT was then determined by the ratio of Gag levels detected from cell culture supernatants to that from the normalized Gag levels from the cell lysate, with WT particle release being set to 100. Results were plotted using GraphPad Prism 6.0. The relative significance between the WT and mutant was determined using an unpaired $t$-test. Cryo-EM analysis of immature particle morphologies of the L43A mutant was performed by producing VLPs from cells and concentrating by ultracentrifugation.

## Cryo-EM analysis of L43A particle morphology

The pN3-HTLV-1-Gag L43A and the HTLV-1 Env expression plasmids were cotransfected into HEK293T/17 cells (ATCC, CRL-11268) using GenJet (version II) at a 10:1 molar ratio as previously described[31,74]. At 48 h after ransfection, cell culture supernatants were harvested and centrifuged at 1,800$g$ for 10 min, followed by passing through a 0.2-μm filter. Particles were then concentrated by ultracentrifugation in a 50.2 Ti rotor at 35,000 r.p.m. for 90 min through an 8% Optiprep cushion. Particle pellets were resuspended in about 200 μl of STE buffer before ultracentrifugation through a 10–30% Optiprep step gradient at 45,000 r.p.m. for 3 h. The particle band was removed from the gradient by puncturing the side of the thin wall ultraclear centrifuge tube (Beckman Coulter) with a hypodermic syringe needle and pelleted in STE buffer at 40,000 r.p.m. for 1 h using an SW55 Ti rotor. The resulting particle pellet was then resuspended in approximately 10 μl of STE buffer and frozen at −80 °C until analyzed by cyro-EM.

Cryo-EM analysis of L43A particles was performed as previously described[31,37,42,74]. Particle samples were thawed on ice and approximately 3.0 μl of purified virus particles were applied to glow-discharged Quantifoil R1.2/1.3 400-mesh holey carbon-coated copper grids. The grids were then manually blotted by a piece of filter paper and plunge-frozen in liquid ethane. The frozen grids were stored in liquid nitrogen until imaging analysis.

Cryo-EM analysis was performed using a Tecnai FEI G2 F30 FEG transmission EM instrument (FEI) at liquid nitrogen temperature operating at 300 kV. Images were recorded at a nominal magnification of ×20,000 at 5–10 μm underfocus using a Gatan 4k-by-4k charge-coupled device camera. At least 250 individual immature particles of L43A mutant were collected.

## Reporting summary

Further information on research design is available in the Nature Portfolio Reporting Summary linked to this article.

## Data availability

The EM density maps and models for the immature HTLV-1 CA-NTD lattice and representative tomograms were deposited to the EM Data Bank under accession codes EMD-17929, EMD-17930, EMD-17931, EMD-17932, EMD-17933, EMD-17934, EMD-17935, EMD-17936, EMD-17937, EMD-17938, EMD-17939, EMD-17940, EMD-17941, EMD-17942 and EMD-17943 and the PDB under accession codes 8PU6, 8PU7, 8PU8, 8PU9, 8PUA, 8PUB, 8PUC, 8PUD, 8PUE, 8PUF, 8PUG and 8PUH. PDB 1QRJ was used as starting model to derive a refined model of the HTLV-1 CA-NTD. The models used to generate Fig. 3 and Extended Data Fig. 3 were PDB 5L93 (HIV-1), PDB 5A9E (RSV), PDB 6HWI (M-PMV) and PDB 6HWW (MLV). The UniProt codes for the protein sequences used for generating Extended Data Fig. 9 were P03345 (GAG_HTL1A), P03346 (GAG_HTLV2), Q0R5R4 (GAG_HTL32), P25058 (GAG_BLVAU), P04585 (POL_HV1H2), P15832 (GAG_HV2D2), P16087 (GAG_FIVPE), P69732 (GAG_EIAVY), P03322 (GAG_RSVP), P0C776 (GAG_ALVA), P07567 (GAG_MPMV), P31622 (GAG_JSRV), P10258 (GAG_MMTVB), P03355 (POL_MLVMS), P10262 (GAG_FLV) and Q9TTC2 (GAG_KORV). Source data are provided with this paper.

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

## Acknowledgements

This work was funded by the Institute of Science and Technology Austria (ISTA) and the Austrian Science Fund (grant P31445 to F.K.M.S.). Access to high-resolution cryo-ET data acquisition at European Molecular Biology Laboratory (EMBL) Heidelberg was supported through the EMBL cryo-EM platform. We thank V.-V. Hodirnau at ISTA and W. Hagen and F. Weis at EMBL Heidelberg for support in cryo-ET data acquisition. This research was also supported by the scientific service units of ISTA through resources provided by Scientific Computing, the Life Science Facility, and the EM Facility. L.M.M. was supported by National Institutes of Health grants R01 GM151775 and R21 DE032878 and by the University of Minnesota Masonic Cancer Center. D.P. was supported by the DOC doctoral fellowship program of the Austrian Academy of Sciences. R.A.D was supported by the National Institute of Allergy and Infectious Diseases (grant R01AI147890). The funders had no role in study design, data collection and analysis, decision to publish or preparation of the manuscript. Specifically, we also want to thank A. Schlögl for computational support and J. Hansen and V. Vogt for critical comments on the manuscript. We also thank the other members of the Schur lab for helpful discussions and experimental advice.

## Author contributions

Project administration, F.K.M.S. Supervision and funding acquisition, L.M.M., R.A.D. and F.K.M.S. Conceptualization, M.O. and F.K.M.S. Methodology, M.O. and F.K.M.S. Investigation, M.O., M.P., D.C., H.Y., A.T., G.P., D.P., R.A.D. and F.K.M.S. Software, M.O. Validation, formal analysis and visualization, M.O., M.P., H.Y. and F.K.M.S. Data curation, M.O. and F.K.M.S. Writing—original draft, M.O. and F.K.M.S. Writing—review and editing, M.O., M.P., D.C., H.Y, A.T., G.P., D.P., L.M.M., R.A.D. and F.K.M.S.

## Competing interests

M.O. is currently an employee of Thermo Fisher Scientific. Thermo Fisher Scientific had no role in study design or experimental aspects of the submitted work and did not provide any financial support for this project. The other authors declare no competing interests.

## Additional information

**Extended data** is available for this paper at https://doi.org/10.1038/s41594-024-01390-8.

**Correspondence and requests for materials** should be addressed to Florian K. M. Schur.

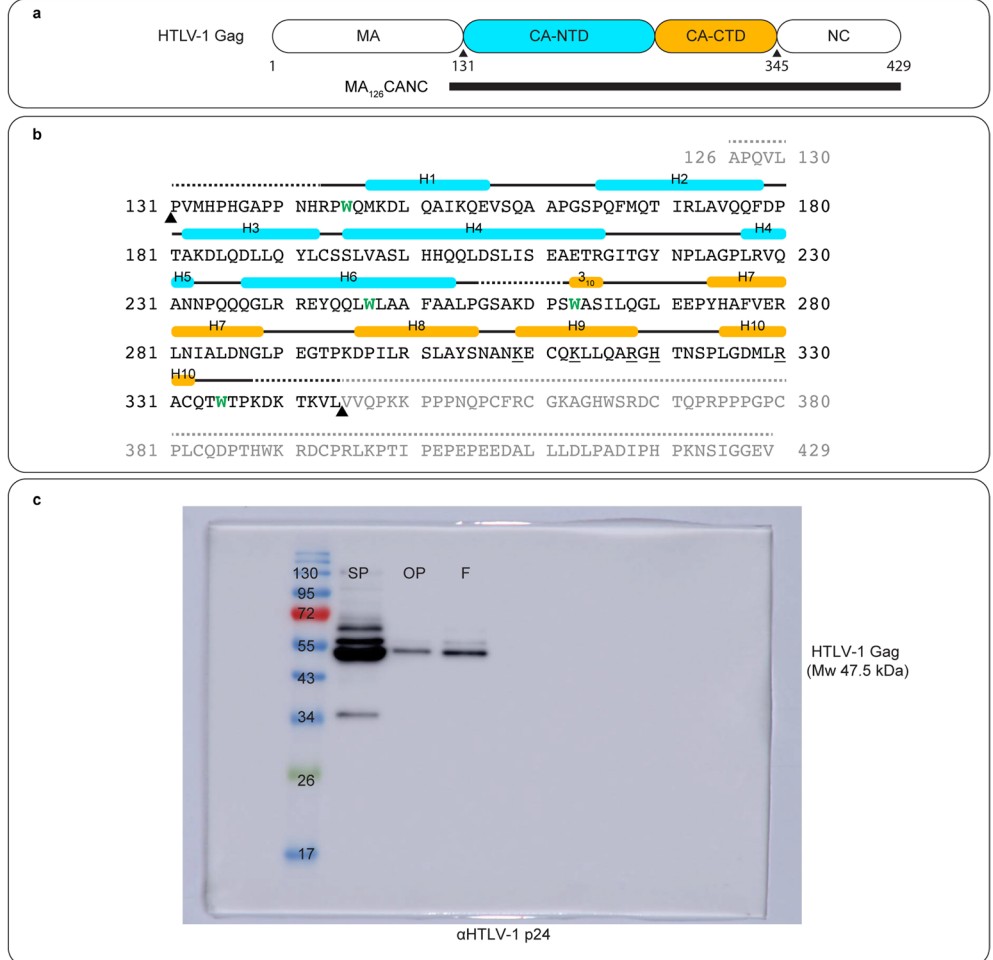

**Extended Data Fig. 1 | HTLV-1 Gag/CA sequence and Gag VLP purification.**
**a)** Schematic depiction of the domain architecture of HTLV-1 Gag and the MA126CANC truncation protein used for *in vitro* assembly and structure determination. Residue numbers and black triangles denote protease cleavage sites. Note the absence of non-canonical Gag domains in HTLV-1. **b)** Sequence of HTLV-1 MA126CANC. The positions of α-helices are shown as cyan and yellow bars. Regions for which structural data is lacking are dashed. Triangles denote protease cleavage sites. Please note that due to the used expression and purification system of our 6xHIS-SUMO construct, an ectopic Serine residue remains at the very N-terminus of the purified protein (not shown in this presented sequence). The underlined residues in Helix 9 and 10 denote positively charged residues which could form potential interactions with the NC-RNA. Tryptophan residues (W) are highlighted in green. **c)** Samples of Optiprep gradient (OP)-purified particles were separated by SDS–polyacrylamide gel electrophoresis. Proteins were visualized by immunoblot using an anti-HTLV-1 p24 antibody. Sucrose pellet (SP), Optiprep bottom (OP), Final Optiprep pellet (F). Positions of molecular mass standards (in kiloDaltons, kDa) are indicated. The molecular weight (MW) of HTLV-1 Gag of 47.5 kDa is annotated. Cell culture production of HTLV-1 Gag VLPs was first established in small scale experiments. The expression was repeated at least 3x with consistent results. The purification blot shown in (c) was performed once to produce the sample in a sufficient quantity to be analysed using cryo-ET.

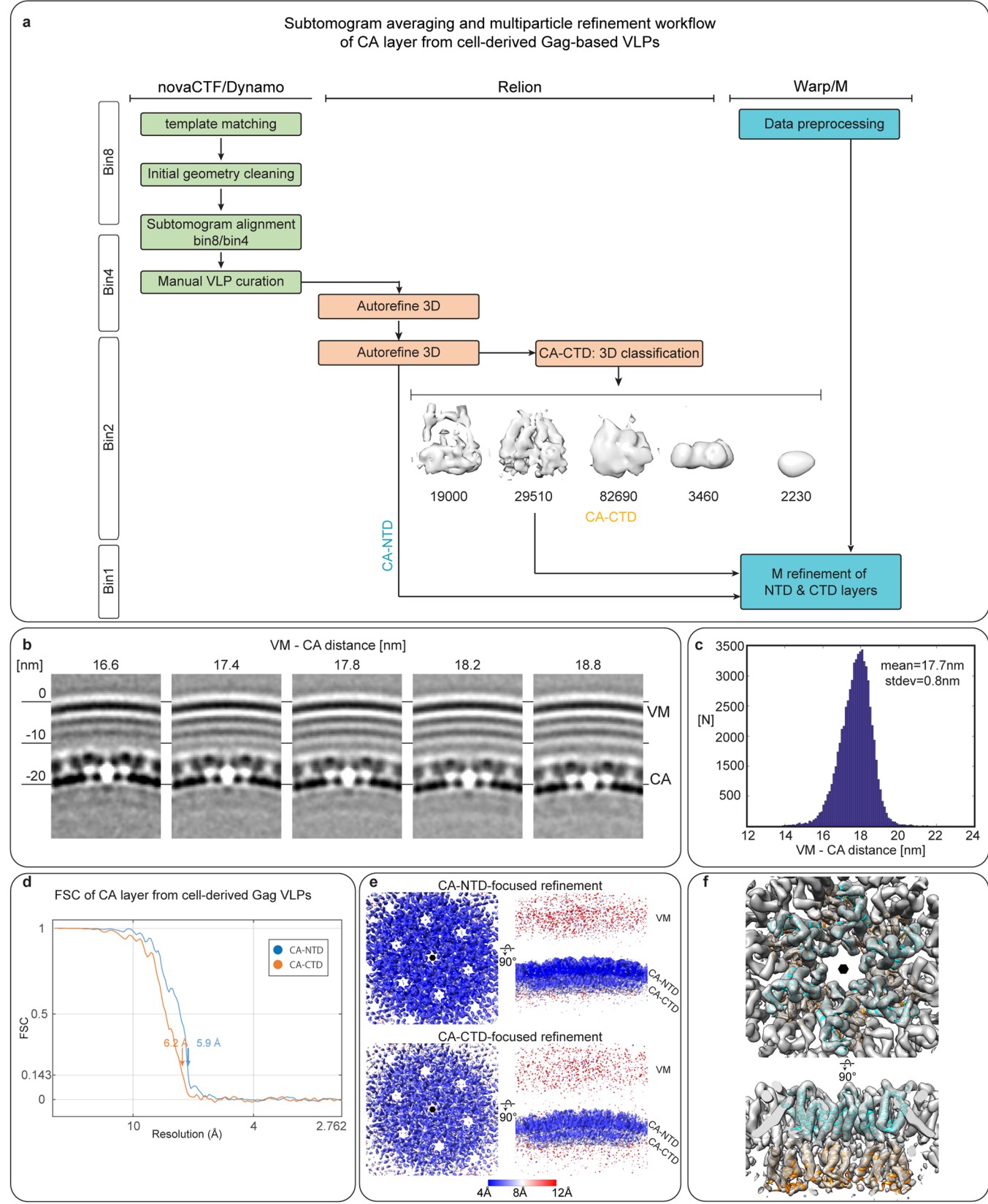

**Extended Data Fig. 2 | See next page for caption.**

**Extended Data Fig. 2 | Image processing of HTLV-1 Gag VLPs. a)** Image processing workflow for the subtomogram averaging and multiparticle refinement of the CA layer from HTLV-1 Gag-based VLPs. See Methods for more details. **b)** XZ-slices through bin4 averages of the immature HTLV-1 Gag lattice, showing the varying distance of the CA-layer with respect to the viral membrane (VM). The distance was measured from the outer VM layer to the electron-lucent layer between CA-NTD and CA-CTD. **c)** Histogram showing the measured distances of the CA-layer to the viral membrane. The distance was determined by re-aligning particles to viral membrane layers. The individual CA-membrane distances then correspond to the mean value + particle Z-shift. N = 51,700. **d)** Fourier-shell correlation (FSC) for the independently aligned CA-NTD (blue line) and CA-CTD layer (red line), showing a resolution of 5.9 Å and 6.2 Å at the 0.143 criterion, respectively. **e)** Local resolution measurements for the two cryo-EM density maps generated by focusing either on the CA-NTD (top) or the CA-CTD (bottom). The color code for the local resolution in Å is shown below. The maps are shown in two views, once as seen from the outside of the VLP (left) and in a side view (right). The viral membrane (VM), the CA-NTD, and CA-CTD are annotated in the side view. The CA hexamer center is indicated with a small black hexagon. **f)** Model of HTLV-1 CA-NTD (cyan) and CA-CTD (orange) rigid-body fitted into a composite map of the immature HTLV-1 CA lattice and shown from two views (as seen from the outside of the VLP on the top, and a side view on the bottom). The density of the CA-NTD is from the cryo-electron microscopy density map refined on the CA-NTD (EMD-17942) and the density for the CA-CTD from EMD-17943). The CA hexamer center is indicated with a small black hexagon.

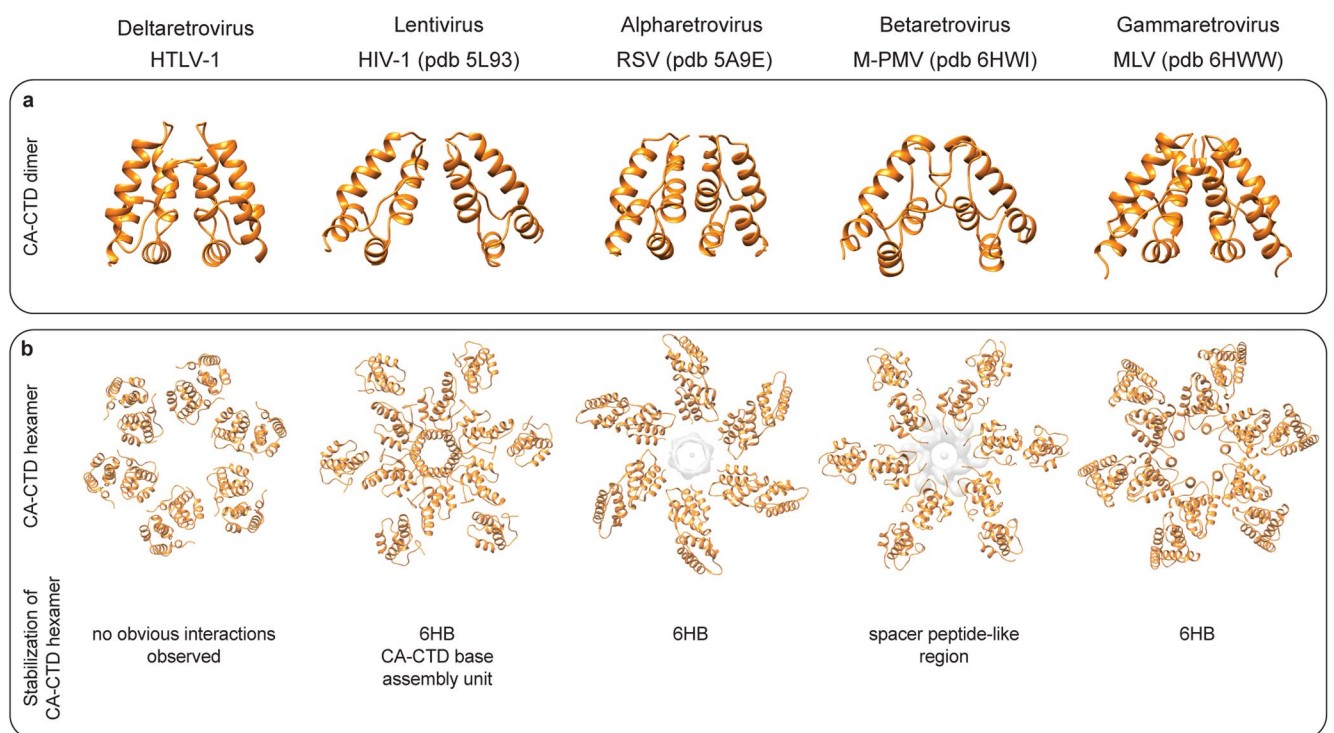

**Extended Data Fig. 3 | CA-CTD interactions in immature retroviral lattices.**
**a)** Comparison of different retroviral immature CA-CTD dimer structures. The pdb accession codes for the different retroviruses are indicated. The same models have been used to generate the data shown in panel (b). **b)** Comparison of the immature CA-CTD hexamer assembly. The main interaction interfaces described previously to stabilize the hexamer are annotated. The CA-CTD assembly unit in HIV-1 is described in[9]. For RSV and M-PMV the electron microscopy densities for the RSV SP and the M-PMV spacer peptide-like region are shown, as for these regions no molecular model has yet been obtained.

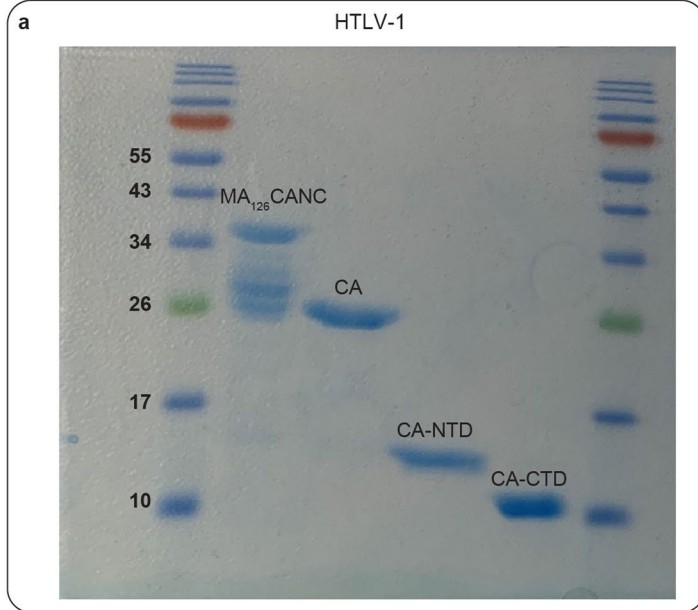

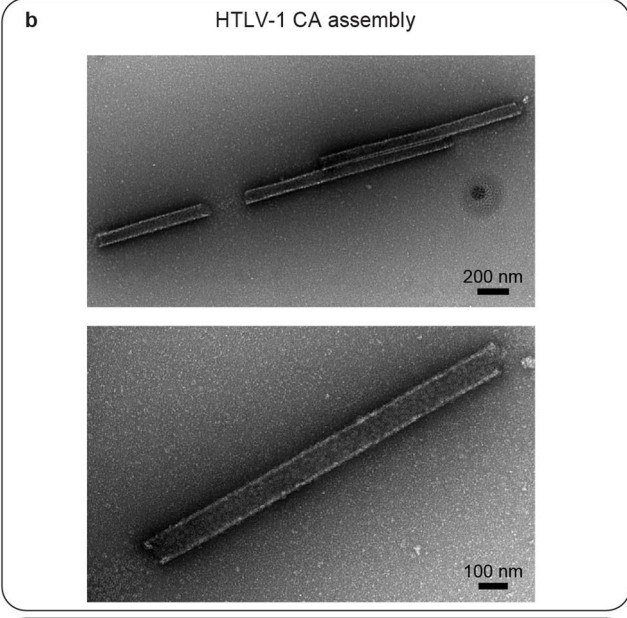

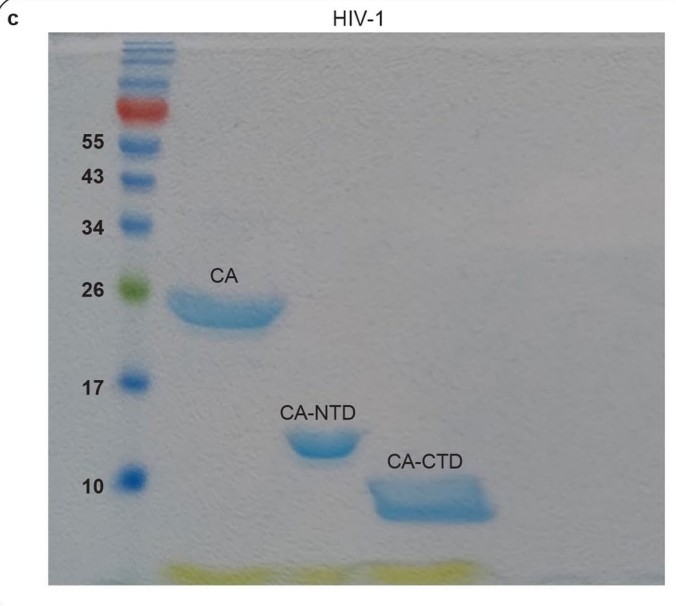

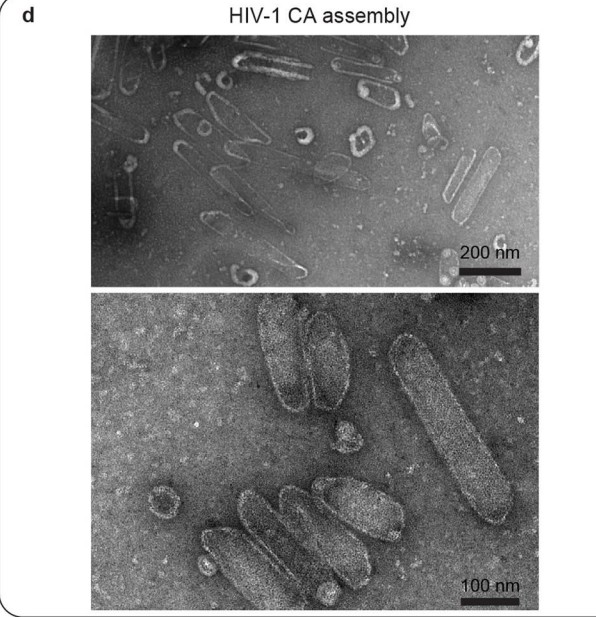

**Extended Data Fig. 4 | Expression and purification of retrovirus CA proteins for *in vitro* experiments. a)** Samples of purified HTLV-1 CA protein variants were separated by SDS–PAGE (15%) and stained via Coomassie brilliant blue. Positions of molecular mass standards (in kiloDaltons, kDa) are indicated. HTLV-1 CA is full-length from residues 131–344 (using Gag-numbering), CA-NTD includes residues 131–258 and CA-CTD includes residues 259–344. **b)** Negative Stain TEM micrographs of HTLV-1 CA tubes. Scale bars are annotated in the figure. **c)** Samples of purified HIV-1 CA protein variants were separated by SDS–PAGE (15%) and stained via Coomassie brilliant blue. Positions of molecular mass standards (kDa) are indicated. HIV-1 CA is full-length from residues 133–363 (using Gag-numbering), CA-NTD includes residues 133–278 and CA-CTD includes residues 279–363. **d)** Negative Stain TEM micrographs of HIV-1 CA assemblies. Scale bars are annotated in the figure. The shown gels and negative stain micrographs are representative for at least 3 repetitions of the same experiment which yielded similar results.

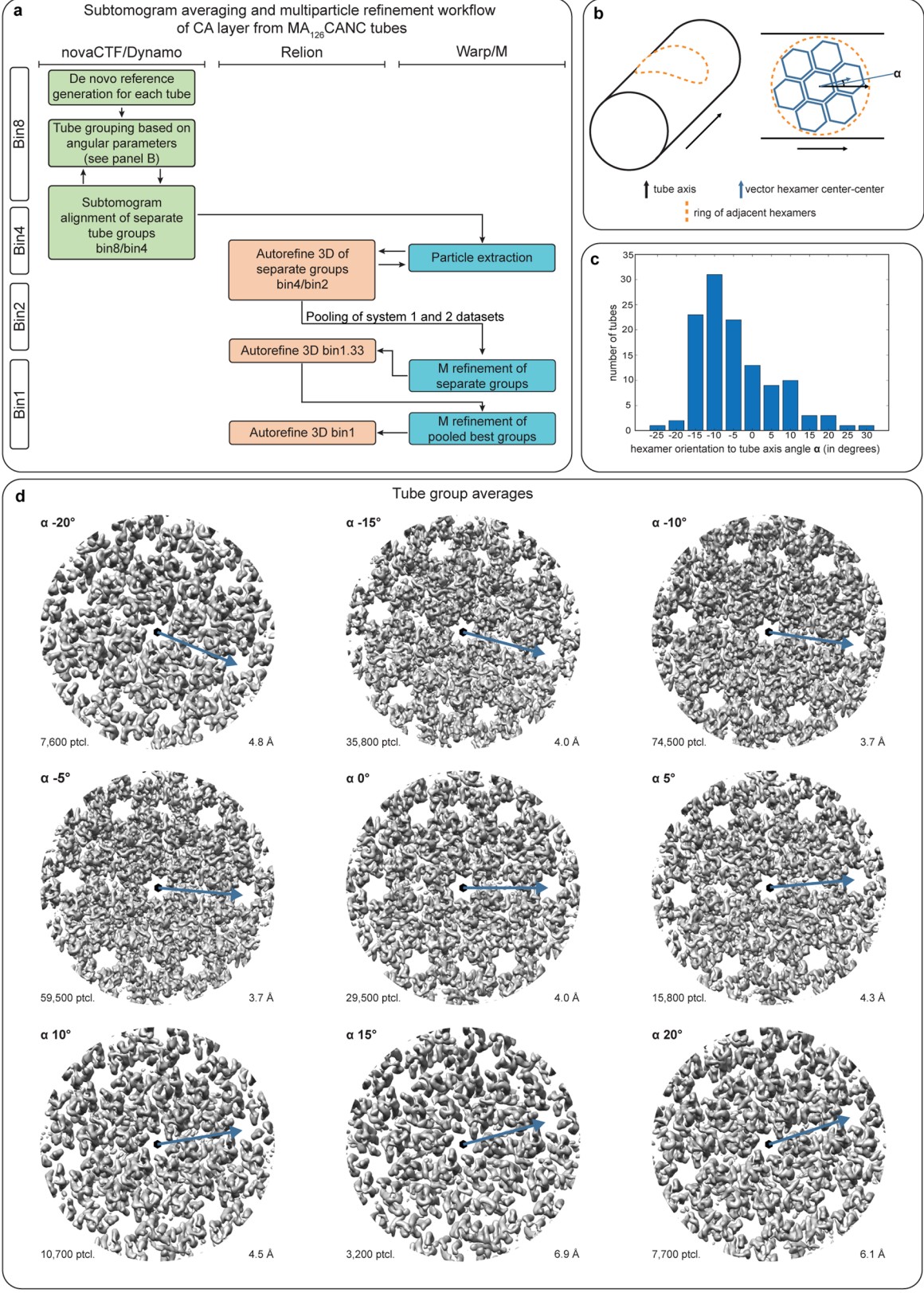

**Extended Data Fig. 5 | Image processing and classification of HTLV-1 MA$_{126}$CANC tubes. a)** Image processing workflow for the subtomogram averaging and multiparticle refinement of the CA-NTD layer from HTLV-1 MA$_{126}$CANC tubes. Please see Materials and Methods for more details. **b)** Classification of tubes based on geometry. Different tubes have different geometries, with varying helical rise and pitch. For improved structure determination, tubes were grouped into classes according to their hexamer orientation (blue arrow) with respect to the tubes axis (black arrow). **c)** Histogram of abundance of different tube groups. **d)** Refined EM-density maps for the individual tube groups. Upper left corner – angle group; lower left corner – number of particles; lower right corner – resolution.

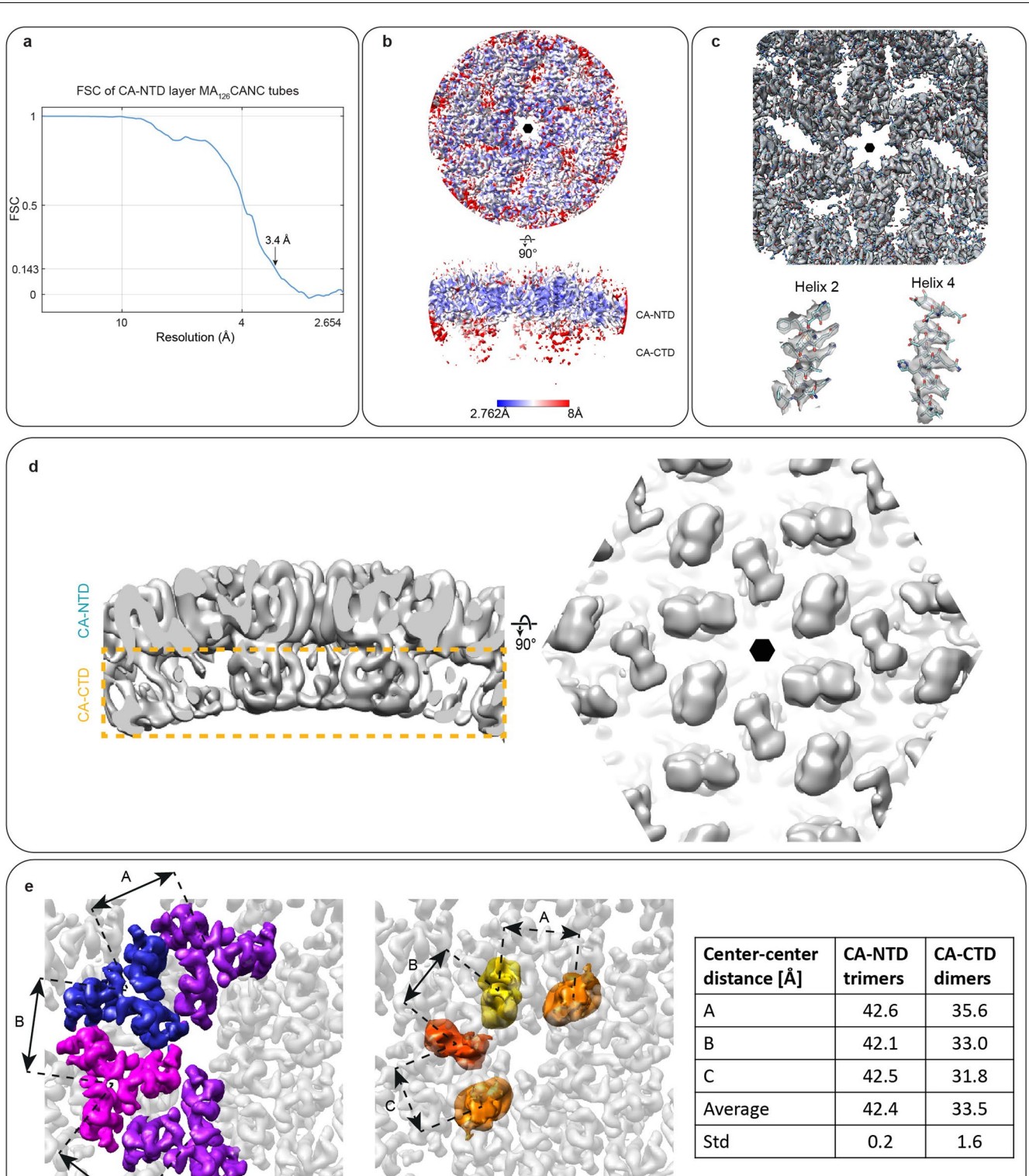

**Extended Data Fig. 6 | High-resolution cryo-ET structure of HTLV-1 MA₁₂₆CANC tubes. a)** FSC for the CA-NTD from MA₁₂₆CANC tubes, showing a resolution of 3.4 Å at the 0.143 criterion. **b)** Local resolution measurements for the cryo-EM density map of the CA layer of HTLV-1 MA₁₂₆CANC tubes tubes. The CA-NTD and CA-CTD are annotated in the side view, highlighting again the weak density for the CA-CTD. The CA hexamer center is indicated with a small black hexagon. **c)** Model of HTLV-1 CA-NTD (cyan) refined into the 3.4 Å cryo-EM density map. The CA hexamer center is indicated with a small black hexagon. Below, representative EM densities for two helices of the CA-NTD are shown with

the respective fit. At this resolution, the helical pitch and densities for larger side chains are visible. **d)** Low-pass-filtered isosurface representation of the subtomogram average of the C2-symmetric CA hexamer from HTLV-1 MA₁₂₆CANC tubes, as seen in a side view (left) or from within the tube, showing the CA-CTD. This panel relates to the Fig. 2d, which shows the same map at the unfiltered 3.4 Å resolution. The orange dashed rectangle indicates the CA-CTD layer, which is not visible at high resolution in Fig. 2d. **e)** Distances between adjacent CA-NTD trimers and CA-CTD dimers.

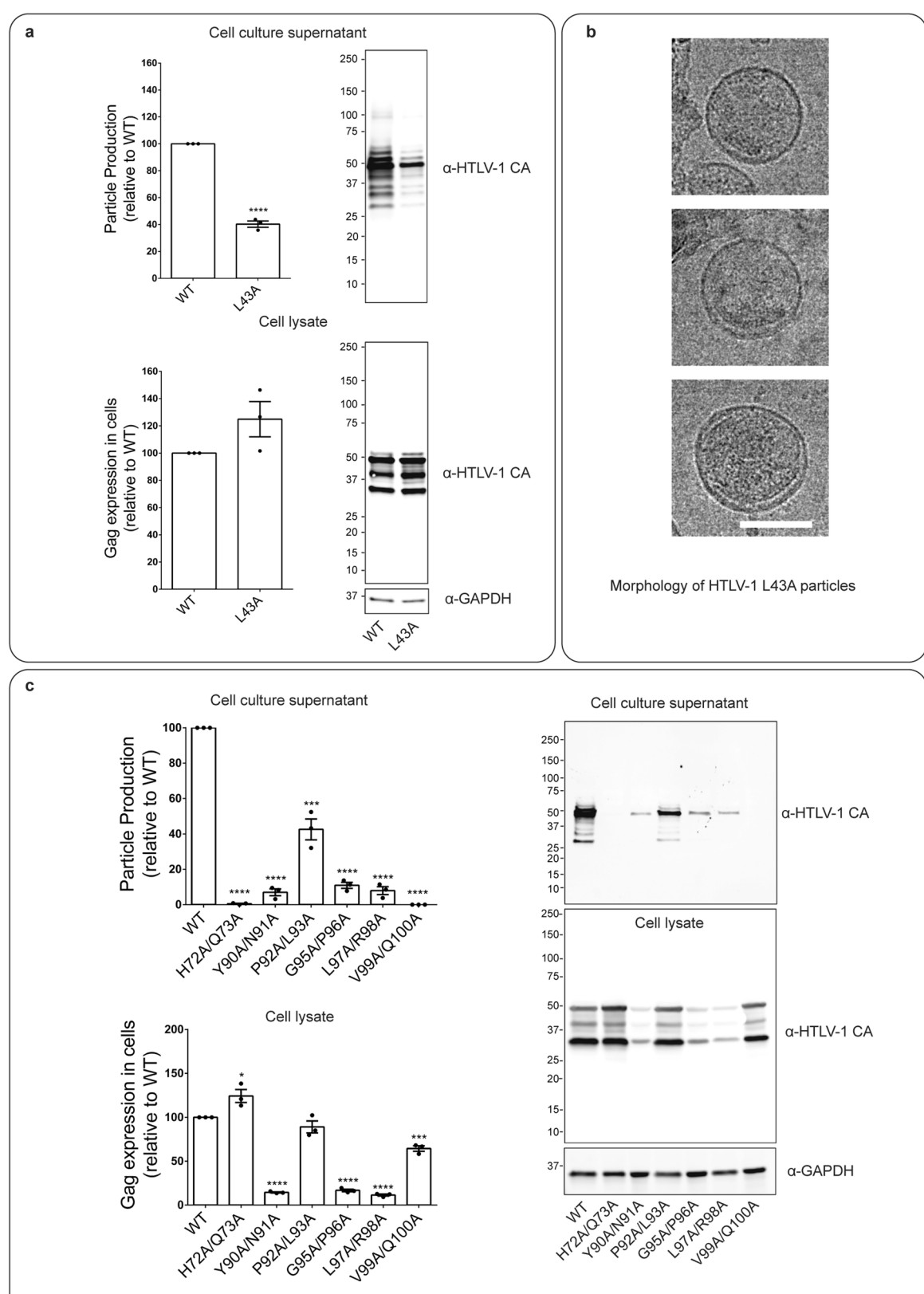

**Extended Data Fig. 7 | See next page for caption.**

**Extended Data Fig. 7 | Mutational analysis of the HTLV-1 CA-NTD interfaces.**
Site-directed mutagenesis of amino acid residues at the CA-NTD interfaces
were analyzed for their ability to produce particles relative to that of the WT.
Transfection of 293 T/17 cells with plasmids expressing the HTLV-1 Gag (WT or
alanine-scanning mutant) was done. Cell culture supernatants and cell lysates
were harvested 48 h post-transfection. **a)** Gag expression in cells and particle
production of L43A mutant. Immunoblot analysis was conducted to determine
particle production relative to that of WT HTLV-1 Gag. Data was collected from
three independent experiments. Error bars represent the standard error of the
mean. Significance relative to WT was determined by using an unpaired two-
sided t-test. ****, $P < 0.0001$. One representative set of immunoblots of the Gag
protein detected from cell culture supernatants and from cell lysates by using
an anti-HTLV p24 (CA) antibody is shown. The locations of the molecular weight
markers (in kilodalton) are indicated along the left side of the immunoblots.

**b)** Representative cryo-EM images showing the morphology of the L43A mutant
particles. Scale bar = 100 nm. The shown examples are representative of at least
250 individual immature L43A mutant particles imaged. **c)** Gag expression in cells
and particle production of the indicated alanine double mutants. Immunoblot
analysis was conducted to determine particle production relative to that of WT
HTLV-1 Gag. Data was collected from three independent experiments. Error
bars represent the standard error of the mean. Significance relative to WT was
determined by using an unpaired two-sided t-test. ****, $P < 0.0001$; ***, $P < 0.001$;
**, $P < 0.01$. One representative set of immunoblots of the Gag protein detected
from cell culture supernatants and from cell lysates by using an anti-HTLV
p24 (CA) antibody is shown. The locations of the molecular weight markers
(in kilodalton) are indicated along the left side of the immunoblots. Detailed
statistical information is given in Supplementary Table 1.

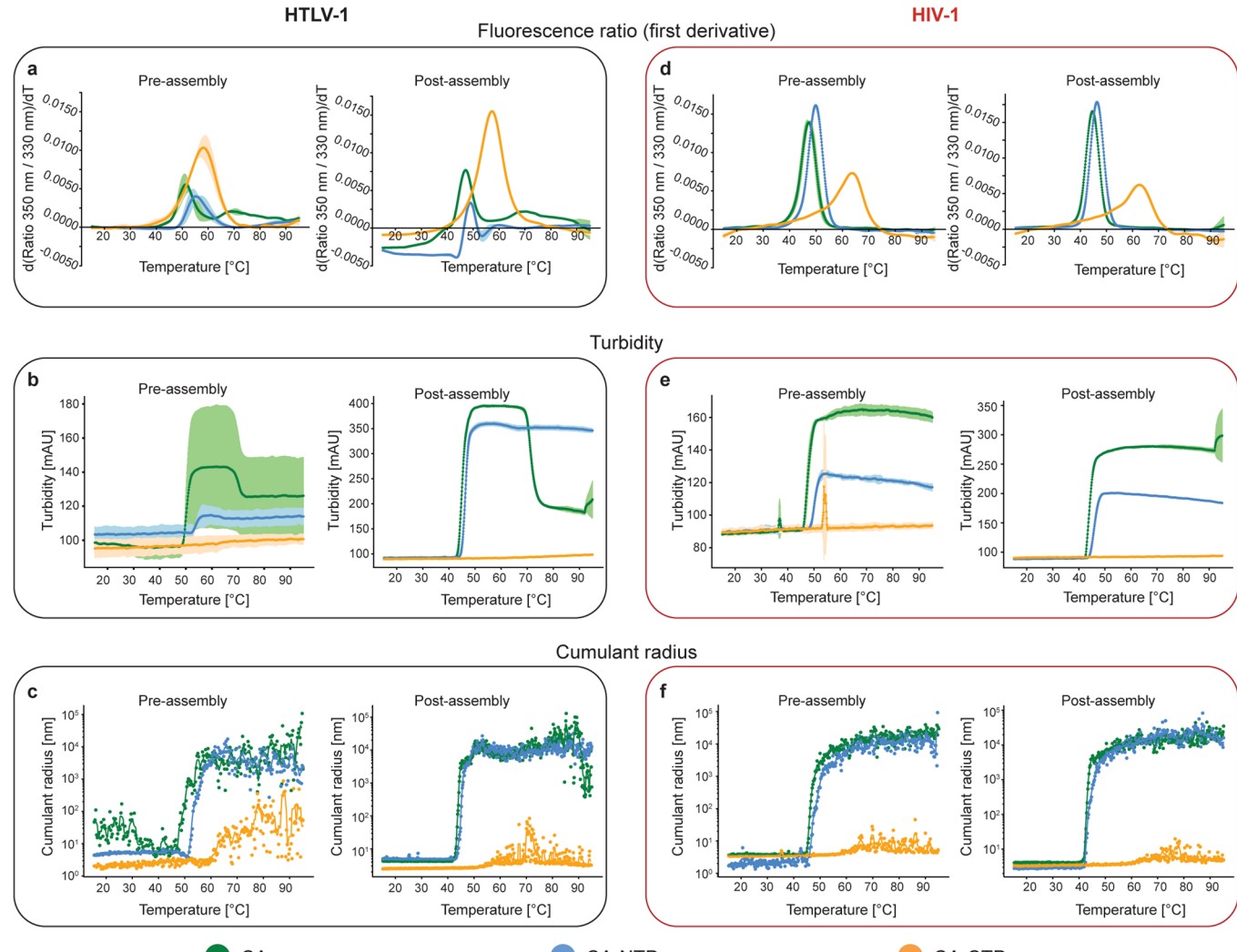

**Extended Data Fig. 8 | Biophysical analysis of HTLV-1 and HIV-1 CA constructs.**
Biophysical analysis of HTLV-1 (panels a-c) and HIV-1 (panels d-f) CA and their
N-terminal and C-terminal domains. Measurements are always performed in
two conditions, termed pre-assembly (left side of respective panel) and post-
assembly (right side of respective panel). **a,d)** First derivative of the fluorescence
ratio over a temperature ramp, measured by nanoDSF, for HTLV-1 (a) and HIV-1
(d) CA (green), CA-NTD (cyan), and CA-CTD (orange) in pre-assembly and post-
assembly conditions. Both full-length HTLV-1 CA and HTLV-1 CA-NTD show a drop
in $T_m$ upon assembly. This drop in $T_m$ might reflect the changed reactivity of CA
and CA-NTD in assembly conditions. Buffer conditions selected for assembly
are chosen to stimulate protein-protein interactions (for example rendering the
proteins to be more assembly reactive). This could make them more dynamic
and also flexible in order to allow forming such interactions. Virus assembly is
not necessarily representing a stable end state, as viruses need to be metastable
to eventually release the viral genome. A $T_m$ drop might therefore reflect the
propensity of the proteins to allow assembly and disassembly. **b,e)** Turbidity
change as a function of temperature, measured by back-reflection, for HTLV-1
(b) and HIV-1 (e) CA, CA-NTD, and CA-CTD in pre-assembly and post-assembly
conditions. $T_m$ for all constructs as well as the turbidity inflection points are
given in Supplementary Table 2. **c,f)** Cumulant radius over a temperature ramp,
measured by DLS, for HTLV-1 (c) and HIV-1 (f) CA, CA-NTD, and CA-CTD in pre-
assembly and post-assembly conditions. CA and CA-NTD show an increase in
cumulant radius coinciding with the observed turbidity increase and respective
melting temperatures. CA-CTD shows no significant cumulant radius increase
across all conditions. The color code is indicated on the bottom of the figure.
The shown data is from one biological replicate measured three times, and
representative for all three replicates performed at least three times. Data for
the fluorescence ratio and turbidity measurements are presented as mean values
+/- standard deviation and for the cumulant radius as mean values with smoothed
curve using a Savitzky-Golay filter with a 5-point window and 0-order polynomial.
Source data is provided alongside the manuscript.

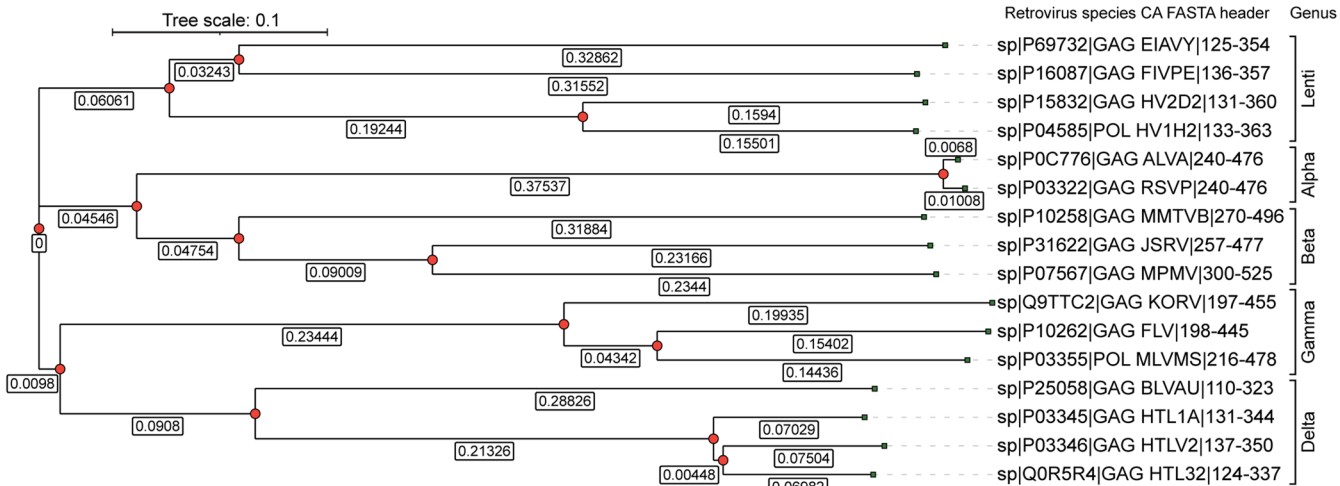

**Extended Data Fig. 9 | Phylogenetic tree of retrovirus CA.** Phylogenetic tree derived from 16 different retrovirus CA sequences, from 5 different retrovirus genera. Multiple sequence alignment (MSA) was performed using Clustal Omega (https://www.ebi.ac.uk/jdispatcher/msa/clustalo) and the phylogenetic tree was generated using the webserver implementation of iTOL (https://itol.embl.de/). The UniProt accession IDs and the CA residues are indicated in the FASTA header on the right side. The boxed numbers indicate the branch length. EIAVY – Equine infectious anemia virus, FIVPE – Feline immunodeficiency virus, HV2 – HIV2, HV1 – HIV1, ALVA – Avian leukosis virus, RSVP – Rous sarcoma Virus, MMTV – Mouse mammary tumor virus, JSRV - Jaagsiekte sheep retrovirus, MPMV – Mason-Pfizer monkey virus, KORV – Koalaretrovirus, FLV – Feline leukemia virus, MLVMS – Murine leukemia virus, BLVAU – Bovine leukemia virus, HTL1A – HTLV1, HTLV2, HTL32 – HTLV3.

# Reporting Summary

## Statistics

For all statistical analyses, confirm that the following items are present in the figure legend, table legend, main text, or Methods section.

| n/a | Confirmed | |
|---|---|---|
| ☐ | ☒ | The exact sample size (*n*) for each experimental group/condition, given as a discrete number and unit of measurement |
| ☐ | ☒ | A statement on whether measurements were taken from distinct samples or whether the same sample was measured repeatedly |
| ☐ | ☒ | The statistical test(s) used AND whether they are one- or two-sided<br>*Only common tests should be described solely by name; describe more complex techniques in the Methods section.* |
| ☒ | ☐ | A description of all covariates tested |
| ☒ | ☐ | A description of any assumptions or corrections, such as tests of normality and adjustment for multiple comparisons |
| ☐ | ☒ | A full description of the statistical parameters including central tendency (e.g. means) or other basic estimates (e.g. regression coefficient) AND variation (e.g. standard deviation) or associated estimates of uncertainty (e.g. confidence intervals) |
| ☐ | ☒ | For null hypothesis testing, the test statistic (e.g. *F*, *t*, *r*) with confidence intervals, effect sizes, degrees of freedom and *P* value noted<br>*Give P values as exact values whenever suitable.* |
| ☒ | ☐ | For Bayesian analysis, information on the choice of priors and Markov chain Monte Carlo settings |
| ☒ | ☐ | For hierarchical and complex designs, identification of the appropriate level for tests and full reporting of outcomes |
| ☒ | ☐ | Estimates of effect sizes (e.g. Cohen's *d*, Pearson's *r*), indicating how they were calculated |

*Our web collection on statistics for biologists contains articles on many of the points above.*

## Software and code

Policy information about availability of computer code

| Data collection | Cryo-EM data acquisition: Digital Micrograph (as included in the Gatan Microscopy software suite) DM 3.31.2360.0 and DM 3.32.2403.0, SerialEM Version 3.8.0 beta 64-bit, FEI AutoCTF software (no version number)<br>NanoDSF: Panta Control (1.5) |
|---|---|
| Data analysis | Cryo-EM data analysis: IMOD (v. 4.9.12), Matlab R2018b, subTOM (no version number available), Dynamo (1.1.133), Warp (1.0.9), ctffind (4.1.10), NovaCTF (no version number available), Relion (3.0.8)<br>NanoDSF: Panta Analysis (1.4.4)<br>Data visualization: UCSF Chimera (v1.17.3), UCSF ChimeraX (v1.7.1, v1.8), GraphPad Prism (6.0), Python (v 3.11.4), Adobe Creative Cloud Illustrator and Premiere Pro (2023) |

For manuscripts utilizing custom algorithms or software that are central to the research but not yet described in published literature, software must be made available to editors and reviewers. We strongly encourage code deposition in a community repository (e.g. GitHub). See the Nature Portfolio guidelines for submitting code & software for further information.

## Data

Policy information about availability of data

All manuscripts must include a data availability statement. This statement should provide the following information, where applicable:
- Accession codes, unique identifiers, or web links for publicly available datasets
- A description of any restrictions on data availability
- For clinical datasets or third party data, please ensure that the statement adheres to our policy

The electron microscopy density maps and models for the immature HTLV-1 CA-NTD lattice and representative tomograms have been deposited in the Electron Microscopy Data Bank (accession codes: EMD-17929, EMD-17930, EMD-17931, EMD-17932, EMD-17933, EMD-17934, EMD-17935, EMD-17936, EMD-17937, EMD-17938, EMD-17939, EMD-17940, EMD-17941, EMD17942, EMD-17943) and the Protein Data Bank (accession codes: 8PU6, 8PU7, 8PU8, 8PU9, 8PUA, 8PUB, 8PUC, 8PUD, 8PUE, 8PUF, 8PUG, 8PUH).
PDB 1QRJ was used as starting model to derive a refined model of the HTLV-1 CA-NTD.
The PDB accession codes for models used to generate Figure 3 and Extended Data Figure 3 are 5L93 (HIV-1), 5A9E (RSV), 6HWI (M-PMV), 6HWW (MLV).
The UniProt codes for the protein sequences used for generating Extended Data Figure 9 are: P03345(GAG_HTL1A), P03346(GAG_HTLV2), Q0R5R4 (GAG_HTL32), P25058 (GAG_BLVAU), P04585 (POL_HV1H2), P15832 (GAG_HV2D2), P16087 (GAG_FIVPE), P69732 (GAG_EIAVY), P03322 (GAG_RSVP), P0C776 (GAG_ALVA), P07567 (GAG_MPMV), P31622 (GAG_JSRV), P10258 (GAG_MMTVB), P03355 (POL_MLVMS), P10262 (GAG_FLV), Q9TTC2 (GAG_KORV).

## Research involving human participants, their data, or biological material

Policy information about studies with human participants or human data. See also policy information about sex, gender (identity/presentation), and sexual orientation and race, ethnicity and racism.

| | |
|---|---|
| Reporting on sex and gender | n/a |
| Reporting on race, ethnicity, or other socially relevant groupings | n/a |
| Population characteristics | n/a |
| Recruitment | n/a |
| Ethics oversight | n/a |

Note that full information on the approval of the study protocol must also be provided in the manuscript.

# Field-specific reporting

Please select the one below that is the best fit for your research. If you are not sure, read the appropriate sections before making your selection.

☒ Life sciences    ☐ Behavioural & social sciences    ☐ Ecological, evolutionary & environmental sciences

For a reference copy of the document with all sections, see nature.com/documents/nr-reporting-summary-flat.pdf

# Life sciences study design

All studies must disclose on these points even when the disclosure is negative.

| | |
|---|---|
| Sample size | Cryo-ET of mammalian cell-derived HTLV-1 Gag VLPs: We used 85 tilt-series for subtomogram averaging, which was sufficient to generate subtomogram averages showing structural features of relevance.<br>Cryo-ET of HTLV-1 MA126CANC VLPs: We used 67 tilt series for subtomogram averaging, which was sufficient to generate subtomogram averages showing structural features of relevance. |
| Data exclusions | Where applicable, data was not excluded. |
| Replication | NanoDSF: At least three different batches of protein were produced and subjected to the biophysical analyses in technical replicates. All attempts at replication were successful.<br>cryo-EM: No replication was performed, as the resolution obtained was high enough for fitting and interpretation of data.<br>Mutagenesis experiments: Data was collected from three independent experiments. All attempts of replication were successful showing the same result. |
| Randomization | Randomization of samples was not relevant in this study since structural characterization was exploratory and no grouping was necessary in order to make relevant comparisons or to draw conclusions. |
| Blinding | Blinding was not relevant in this study since macromolecular structures and assembly details were unknown to any researcher involved. |

# Behavioural & social sciences study design

All studies must disclose on these points even when the disclosure is negative.

| | |
|---|---|
| Study description | *Briefly describe the study type including whether data are quantitative, qualitative, or mixed-methods (e.g. qualitative cross-sectional, quantitative experimental, mixed-methods case study).* |
| Research sample | *State the research sample (e.g. Harvard university undergraduates, villagers in rural India) and provide relevant demographic information (e.g. age, sex) and indicate whether the sample is representative. Provide a rationale for the study sample chosen. For studies involving existing datasets, please describe the dataset and source.* |
| Sampling strategy | *Describe the sampling procedure (e.g. random, snowball, stratified, convenience). Describe the statistical methods that were used to predetermine sample size OR if no sample-size calculation was performed, describe how sample sizes were chosen and provide a rationale for why these sample sizes are sufficient. For qualitative data, please indicate whether data saturation was considered, and what criteria were used to decide that no further sampling was needed.* |
| Data collection | *Provide details about the data collection procedure, including the instruments or devices used to record the data (e.g. pen and paper, computer, eye tracker, video or audio equipment) whether anyone was present besides the participant(s) and the researcher, and whether the researcher was blind to experimental condition and/or the study hypothesis during data collection.* |
| Timing | *Indicate the start and stop dates of data collection. If there is a gap between collection periods, state the dates for each sample cohort.* |
| Data exclusions | *If no data were excluded from the analyses, state so OR if data were excluded, provide the exact number of exclusions and the rationale behind them, indicating whether exclusion criteria were pre-established.* |
| Non-participation | *State how many participants dropped out/declined participation and the reason(s) given OR provide response rate OR state that no participants dropped out/declined participation.* |
| Randomization | *If participants were not allocated into experimental groups, state so OR describe how participants were allocated to groups, and if allocation was not random, describe how covariates were controlled.* |

# Ecological, evolutionary & environmental sciences study design

All studies must disclose on these points even when the disclosure is negative.

| | |
|---|---|
| Study description | *Briefly describe the study. For quantitative data include treatment factors and interactions, design structure (e.g. factorial, nested, hierarchical), nature and number of experimental units and replicates.* |
| Research sample | *Describe the research sample (e.g. a group of tagged Passer domesticus, all Stenocereus thurberi within Organ Pipe Cactus National Monument), and provide a rationale for the sample choice. When relevant, describe the organism taxa, source, sex, age range and any manipulations. State what population the sample is meant to represent when applicable. For studies involving existing datasets, describe the data and its source.* |
| Sampling strategy | *Note the sampling procedure. Describe the statistical methods that were used to predetermine sample size OR if no sample-size calculation was performed, describe how sample sizes were chosen and provide a rationale for why these sample sizes are sufficient.* |
| Data collection | *Describe the data collection procedure, including who recorded the data and how.* |
| Timing and spatial scale | *Indicate the start and stop dates of data collection, noting the frequency and periodicity of sampling and providing a rationale for these choices. If there is a gap between collection periods, state the dates for each sample cohort. Specify the spatial scale from which the data are taken* |
| Data exclusions | *If no data were excluded from the analyses, state so OR if data were excluded, describe the exclusions and the rationale behind them, indicating whether exclusion criteria were pre-established.* |
| Reproducibility | *Describe the measures taken to verify the reproducibility of experimental findings. For each experiment, note whether any attempts to repeat the experiment failed OR state that all attempts to repeat the experiment were successful.* |
| Randomization | *Describe how samples/organisms/participants were allocated into groups. If allocation was not random, describe how covariates were controlled. If this is not relevant to your study, explain why.* |
| Blinding | *Describe the extent of blinding used during data acquisition and analysis. If blinding was not possible, describe why OR explain why blinding was not relevant to your study.* |

Did the study involve field work? ☐ Yes ☐ No

## Field work, collection and transport

| Field conditions | *Describe the study conditions for field work, providing relevant parameters (e.g. temperature, rainfall).* |
| Location | *State the location of the sampling or experiment, providing relevant parameters (e.g. latitude and longitude, elevation, water depth).* |
| Access & import/export | *Describe the efforts you have made to access habitats and to collect and import/export your samples in a responsible manner and in compliance with local, national and international laws, noting any permits that were obtained (give the name of the issuing authority, the date of issue, and any identifying information).* |
| Disturbance | *Describe any disturbance caused by the study and how it was minimized.* |

# Reporting for specific materials, systems and methods

We require information from authors about some types of materials, experimental systems and methods used in many studies. Here, indicate whether each material, system or method listed is relevant to your study. If you are not sure if a list item applies to your research, read the appropriate section before selecting a response.

### Materials & experimental systems

| n/a | Involved in the study |
|-----|-----------------------|
| ☐ | ☒ Antibodies |
| ☐ | ☒ Eukaryotic cell lines |
| ☒ | ☐ Palaeontology and archaeology |
| ☒ | ☐ Animals and other organisms |
| ☒ | ☐ Clinical data |
| ☒ | ☐ Dual use research of concern |
| ☒ | ☐ Plants |

### Methods

| n/a | Involved in the study |
|-----|-----------------------|
| ☒ | ☐ ChIP-seq |
| ☒ | ☐ Flow cytometry |
| ☒ | ☐ MRI-based neuroimaging |

## Antibodies

| Antibodies used | monoclonal anti-HTLV p24 (CA) antibody (Catalogue #: sc-53891; Santa CruzBiotechnology, Dallas, TX).<br>Extended Data Figure 1c: The dilution used for anti-HTLV p24 (CA) was 1:500<br>Extended Data Figure 7: for supernatant the anti-HTLV p24 (CA) dilution was 1:2000. For cell lysate the anti-HTLV p24 (CA) dilution was 1:1500. |
| Validation | Relevant information on the antibody and its use in other scientific publications is given on the manufacturer's webpage https://www.scbt.com/p/htlv-1-p24-antibody-6g9 |

## Eukaryotic cell lines

Policy information about cell lines and Sex and Gender in Research

| Cell line source(s) | HEK293T and HEK293T/17 cells were ordered from ATCC (ATCC-CRL-3216 and CRL-11268). |
| Authentication | Cell lines were not authenticated. |
| Mycoplasma contamination | Cells were tested negative for Mycoplasma infection |
| Commonly misidentified lines<br>(See ICLAC register) | no commonly misidentified cell lines were used in this study |

## Palaeontology and Archaeology

| Specimen provenance | *Provide provenance information for specimens and describe permits that were obtained for the work (including the name of the issuing authority, the date of issue, and any identifying information). Permits should encompass collection and, where applicable, export.* |
| Specimen deposition | *Indicate where the specimens have been deposited to permit free access by other researchers.* |
| Dating methods | *If new dates are provided, describe how they were obtained (e.g. collection, storage, sample pretreatment and measurement), where* |

| Dating methods | *they were obtained (i.e. lab name), the calibration program and the protocol for quality assurance OR state that no new dates are provided.* |

☐ Tick this box to confirm that the raw and calibrated dates are available in the paper or in Supplementary Information.

| Ethics oversight | *Identify the organization(s) that approved or provided guidance on the study protocol, OR state that no ethical approval or guidance was required and explain why not.* |

Note that full information on the approval of the study protocol must also be provided in the manuscript.

# Animals and other research organisms

Policy information about studies involving animals; ARRIVE guidelines recommended for reporting animal research, and Sex and Gender in Research

| Laboratory animals | *For laboratory animals, report species, strain and age OR state that the study did not involve laboratory animals.* |
| Wild animals | *Provide details on animals observed in or captured in the field; report species and age where possible. Describe how animals were caught and transported and what happened to captive animals after the study (if killed, explain why and describe method; if released, say where and when) OR state that the study did not involve wild animals.* |
| Reporting on sex | *Indicate if findings apply to only one sex; describe whether sex was considered in study design, methods used for assigning sex. Provide data disaggregated for sex where this information has been collected in the source data as appropriate; provide overall numbers in this Reporting Summary. Please state if this information has not been collected. Report sex-based analyses where performed, justify reasons for lack of sex-based analysis.* |
| Field-collected samples | *For laboratory work with field-collected samples, describe all relevant parameters such as housing, maintenance, temperature, photoperiod and end-of-experiment protocol OR state that the study did not involve samples collected from the field.* |
| Ethics oversight | *Identify the organization(s) that approved or provided guidance on the study protocol, OR state that no ethical approval or guidance was required and explain why not.* |

Note that full information on the approval of the study protocol must also be provided in the manuscript.

# Clinical data

Policy information about clinical studies
All manuscripts should comply with the ICMJE guidelines for publication of clinical research and a completed CONSORT checklist must be included with all submissions.

| Clinical trial registration | *Provide the trial registration number from ClinicalTrials.gov or an equivalent agency.* |
| Study protocol | *Note where the full trial protocol can be accessed OR if not available, explain why.* |
| Data collection | *Describe the settings and locales of data collection, noting the time periods of recruitment and data collection.* |
| Outcomes | *Describe how you pre-defined primary and secondary outcome measures and how you assessed these measures.* |

# Dual use research of concern

Policy information about dual use research of concern

## Hazards

Could the accidental, deliberate or reckless misuse of agents or technologies generated in the work, or the application of information presented in the manuscript, pose a threat to:

| No | Yes | |
|----|-----|---|
| ☒ | ☐ | Public health |
| ☒ | ☐ | National security |
| ☒ | ☐ | Crops and/or livestock |
| ☒ | ☐ | Ecosystems |
| ☒ | ☐ | Any other significant area |

## Experiments of concern

Does the work involve any of these experiments of concern:

| No | Yes | |
|----|-----|--|
| ☒ | ☐ | Demonstrate how to render a vaccine ineffective |
| ☒ | ☐ | Confer resistance to therapeutically useful antibiotics or antiviral agents |
| ☒ | ☐ | Enhance the virulence of a pathogen or render a nonpathogen virulent |
| ☒ | ☐ | Increase transmissibility of a pathogen |
| ☒ | ☐ | Alter the host range of a pathogen |
| ☒ | ☐ | Enable evasion of diagnostic/detection modalities |
| ☒ | ☐ | Enable the weaponization of a biological agent or toxin |
| ☒ | ☐ | Any other potentially harmful combination of experiments and agents |

## Plants

| | |
|--|--|
| Seed stocks | *Report on the source of all seed stocks or other plant material used. If applicable, state the seed stock centre and catalogue number. If plant specimens were collected from the field, describe the collection location, date and sampling procedures.* |
| Novel plant genotypes | *Describe the methods by which all novel plant genotypes were produced. This includes those generated by transgenic approaches, gene editing, chemical/radiation-based mutagenesis and hybridization. For transgenic lines, describe the transformation method, the number of independent lines analyzed and the generation upon which experiments were performed. For gene-edited lines, describe the editor used, the endogenous sequence targeted for editing, the targeting guide RNA sequence (if applicable) and how the editor was applied.* |
| Authentication | *Describe any authentication procedures for each seed stock used or novel genotype generated. Describe any experiments used to assess the effect of a mutation and, where applicable, how potential secondary effects (e.g. second site T-DNA insertions, mosiacism, off-target gene editing) were examined.* |

## ChIP-seq

### Data deposition

☐ Confirm that both raw and final processed data have been deposited in a public database such as GEO.

☐ Confirm that you have deposited or provided access to graph files (e.g. BED files) for the called peaks.

| | |
|--|--|
| Data access links<br>*May remain private before publication.* | *For "Initial submission" or "Revised version" documents, provide reviewer access links. For your "Final submission" document, provide a link to the deposited data.* |
| Files in database submission | *Provide a list of all files available in the database submission.* |
| Genome browser session<br>(e.g. UCSC) | *Provide a link to an anonymized genome browser session for "Initial submission" and "Revised version" documents only, to enable peer review. Write "no longer applicable" for "Final submission" documents.* |

### Methodology

| | |
|--|--|
| Replicates | *Describe the experimental replicates, specifying number, type and replicate agreement.* |
| Sequencing depth | *Describe the sequencing depth for each experiment, providing the total number of reads, uniquely mapped reads, length of reads and whether they were paired- or single-end.* |
| Antibodies | *Describe the antibodies used for the ChIP-seq experiments; as applicable, provide supplier name, catalog number, clone name, and lot number.* |
| Peak calling parameters | *Specify the command line program and parameters used for read mapping and peak calling, including the ChIP, control and index files used.* |
| Data quality | *Describe the methods used to ensure data quality in full detail, including how many peaks are at FDR 5% and above 5-fold enrichment.* |
| Software | *Describe the software used to collect and analyze the ChIP-seq data. For custom code that has been deposited into a community repository, provide accession details.* |

# Flow Cytometry

## Plots

Confirm that:

- ☐ The axis labels state the marker and fluorochrome used (e.g. CD4-FITC).
- ☐ The axis scales are clearly visible. Include numbers along axes only for bottom left plot of group (a 'group' is an analysis of identical markers).
- ☐ All plots are contour plots with outliers or pseudocolor plots.
- ☐ A numerical value for number of cells or percentage (with statistics) is provided.

## Methodology

| | |
|---|---|
| Sample preparation | *Describe the sample preparation, detailing the biological source of the cells and any tissue processing steps used.* |
| Instrument | *Identify the instrument used for data collection, specifying make and model number.* |
| Software | *Describe the software used to collect and analyze the flow cytometry data. For custom code that has been deposited into a community repository, provide accession details.* |
| Cell population abundance | *Describe the abundance of the relevant cell populations within post-sort fractions, providing details on the purity of the samples and how it was determined.* |
| Gating strategy | *Describe the gating strategy used for all relevant experiments, specifying the preliminary FSC/SSC gates of the starting cell population, indicating where boundaries between "positive" and "negative" staining cell populations are defined.* |

- ☐ Tick this box to confirm that a figure exemplifying the gating strategy is provided in the Supplementary Information.

# Magnetic resonance imaging

## Experimental design

| | |
|---|---|
| Design type | *Indicate task or resting state; event-related or block design.* |
| Design specifications | *Specify the number of blocks, trials or experimental units per session and/or subject, and specify the length of each trial or block (if trials are blocked) and interval between trials.* |
| Behavioral performance measures | *State number and/or type of variables recorded (e.g. correct button press, response time) and what statistics were used to establish that the subjects were performing the task as expected (e.g. mean, range, and/or standard deviation across subjects).* |

## Acquisition

| | |
|---|---|
| Imaging type(s) | *Specify: functional, structural, diffusion, perfusion.* |
| Field strength | *Specify in Tesla* |
| Sequence & imaging parameters | *Specify the pulse sequence type (gradient echo, spin echo, etc.), imaging type (EPI, spiral, etc.), field of view, matrix size, slice thickness, orientation and TE/TR/flip angle.* |
| Area of acquisition | *State whether a whole brain scan was used OR define the area of acquisition, describing how the region was determined.* |

Diffusion MRI    ☐ Used    ☒ Not used

## Preprocessing

| | |
|---|---|
| Preprocessing software | *Provide detail on software version and revision number and on specific parameters (model/functions, brain extraction, segmentation, smoothing kernel size, etc.).* |
| Normalization | *If data were normalized/standardized, describe the approach(es): specify linear or non-linear and define image types used for transformation OR indicate that data were not normalized and explain rationale for lack of normalization.* |
| Normalization template | *Describe the template used for normalization/transformation, specifying subject space or group standardized space (e.g. original Talairach, MNI305, ICBM152) OR indicate that the data were not normalized.* |
| Noise and artifact removal | *Describe your procedure(s) for artifact and structured noise removal, specifying motion parameters, tissue signals and physiological signals (heart rate, respiration).* |

| Volume censoring | *Define your software and/or method and criteria for volume censoring, and state the extent of such censoring.* |

## Statistical modeling & inference

| Model type and settings | *Specify type (mass univariate, multivariate, RSA, predictive, etc.) and describe essential details of the model at the first and second levels (e.g. fixed, random or mixed effects; drift or auto-correlation).* |

| Effect(s) tested | *Define precise effect in terms of the task or stimulus conditions instead of psychological concepts and indicate whether ANOVA or factorial designs were used.* |

Specify type of analysis: ☐ Whole brain ☐ ROI-based ☐ Both

| Statistic type for inference | *Specify voxel-wise or cluster-wise and report all relevant parameters for cluster-wise methods.* |

(See Eklund et al. 2016)

| Correction | *Describe the type of correction and how it is obtained for multiple comparisons (e.g. FWE, FDR, permutation or Monte Carlo).* |

## Models & analysis

n/a | Involved in the study

☒ ☐ Functional and/or effective connectivity

☒ ☐ Graph analysis

☒ ☐ Multivariate modeling or predictive analysis

