## [Peer Review File · Nature Structural & Molecular Biology]

Peer Review Information

Manuscript Title: Unconventional stabilization of the human T-cell leukemia virus type 1 immature Gag lattice

Corresponding author name(s): Florian Schur

Reviewer Comments & Decisions:

Decision Letter, initial version:

Message: 22nd Sep 2023

Dear Dr Schur,

Thank you again for submitting your manuscript "Unconventional stabilization of the human T-cell leukemia virus type 1 immature Gag lattice". I apologize for the delay in responding, which resulted from the difficulty in obtaining suitable referee reports. Nevertheless, we now have comments (below) from the 3 reviewers who evaluated your paper. In light of those reports, we remain interested in your study and would like to see your response to the comments of the referees, in the form of a revised manuscript.

You will see that while reviewers appreciate the results, they raise several concerns which will need to be addressed in a revision. Specifically, we would ask that in line with reviewer's #1 comments, you discuss further the evolutionary relationship within the described viruses. Related to that, we strongly believe that addressing reviewer's #2 comments will significantly strengthen the manuscript, especially those pertaining to further strengthening biological relevance of the findings, and the questions about generalisability. If these cannot be addressed experimentally, pitfalls should be discussed.

Please be sure to address/respond to all concerns of the referees in full in a point-by-point response and highlight all changes in the revised manuscript text file.

We appreciate the requested revisions are extensive. We thus expect to see your revised manuscript within 6 months. If you cannot send it within this time, please let us know. We will be happy to consider your revision as long as nothing similar has been accepted for publication at NSMB or published elsewhere. Should your manuscript be substantially

delayed without notifying us in advance and your article is eventually published, the received date would be that of the revised, not the original, version.

Reporting Summary:

When submitting the revised version of your manuscript, please pay close attention to our [href="https://www.nature.com/nature-portfolio/editorial-policies/image-integrity">Digital Image Integrity Guidelines](https://www.nature.com/nature-portfolio/editorial-policies/image-integrity). and to the following points below:

Please note that all key data shown in the main figures as cropped gels or blots must be presented in uncropped form, with molecular weight markers. These data can be aggregated into a single supplementary figure. While these data can be displayed in a relatively informal style, they must refer back to the relevant figures.

SOURCE DATA: we request that authors provide, in tabular form, the data underlying the graphical representations used in figures. This is to further increase transparency in data reporting, as detailed in this editorial (<http://www.nature.com/nsmb/journal/v22/n10/full/nsmb.3110.html>). Spreadsheets can be submitted in excel format. Only one (1) file per figure is permitted; thus, for multi-paneled figures, the source data for each panel should be clearly labeled in the Excel file; alternately the data can be provided as multiple, clearly labeled sheets in an Excel file. When submitting files, the title field should indicate which figure the source data pertains to.

We require deposition of coordinates (and, in the case of crystal structures, structure factors) into the Protein Data Bank with the designation of immediate release upon publication (HPUB). Electron microscopy-derived density maps and coordinate data must be deposited in EMDB and released upon publication. Deposition and immediate release of NMR chemical shift assignments are highly encouraged. Deposition of deep sequencing and microarray data is mandatory, and the datasets must be released prior to or upon publication. To avoid delays in publication, dataset accession numbers must be supplied with the final accepted manuscript and appropriate release dates must be indicated at the galley proof stage. Please find the complete NRG policies on data availability at <http://www.nature.com/authors/policies/availability.html>.

Nature Structural & Molecular Biology is committed to improving transparency in authorship. As part of our efforts in this direction, we are now requesting that all authors identified as 'corresponding author' on published papers create and link their Open Researcher and Contributor Identifier (ORCID) with their account on the Manuscript Tracking System (MTS), prior to acceptance. This applies to primary research papers only. ORCID helps the scientific community achieve unambiguous attribution of all scholarly contributions. You can create and link your ORCID from the home page of the MTS by clicking on 'Modify my Springer Nature account'. For more information please visit please visit www.springernature.com/orcid.

[Redacted]

Sincerely,
Kat

Katarzyna Ciazynska
(she/her)
Associate Editor
Nature Structural & Molecular Biology
<https://orcid.org/0000-0002-9899-2428>

Referee expertise:

Referee #1: retroviruses, cryo-ET

Referee #2: retroviruses, cryo-EM

Referee #3: retroviral assembly

Reviewers' Comments:

Reviewer #1:

Remarks to the Author:

In this manuscript Obr and others present detailed and high-quality structural evidence on HTLV-1 immature lattice. These findings explain sui generis characteristics of HTLV-1 Gag protein organization, not found in other retroviruses. While these unique features were previously known through virology, fluorescence microscopy and cryoEM/ET data, there was no data at high-resolution available to mechanistically explain how these qualities were achieved until this manuscript. The data processing approach is on par with the most advanced data processing workflows of the field, which is corroborated by the resolution achieved in the presented subtomogram averaging maps. The quality of data presentation is good, but can be improved by the suggestions listed below in this review. The conclusions are appropriate and based on the structural findings produced by Obr and colleagues. Text is clearly written, and contextual information based on previous literature is well introduced. I recommend this manuscript to be accepted with minor revisions summarized below.

Figure 1:

Panel C shows the radial density plot of Gag lattice. The x axis shows distance in nm from the center of the box (I assume). To be more informative, the distance should state the distance from a known feature, such as the viral membrane. This would also facilitate comparison with similar radial density profiles from the literature. As it stands, the distance from the center of the box makes such interpretations and comparisons more difficult than it should be.

The same can be said from panel B in Figure 2 (in this case, the distance can be represented in relationship to the CA NTD layer/outer layer of tube).

Figure 1 legend:

In the description of E), replace "dashed yellow hexamer" for "dashed yellow hexagon".

Also remove the reference to "annotated as described in the main text". There is no description in main text, instead, the description is on the legend of panels D-F. In

description of G) add mention to the NTD helices involved in the trimeric interface (helices 4 and 5). This will also make the legend consistent to H) description.

Figure 2:

Panel A should have a box showing general placement for the radial density plot, similar to the one showed in Figure 1A.

Panel B suffers from the same issue as described previously for Figure 1C. Distances in x axis should be represented in relationship to the CA NTD layer/outer layer of tube.

Panel C, right volume is not informative. Please change cropping of the lattice to show a side view showing the coloured volumes on the left/top volume. As it is, the grey density obscures them and prevents making any relationship between the monomers in side view.

Panel D depicting NTD hexamer interactions should be shifted to show more context between I and I'. M17 in I seems to be reaching to I', but the way the box is placed prevents making any conclusions or analysis.

Discussion, line 193-196

Obr and colleagues introduce an interesting correlation between HTLV and MLV CA-NTD arrangements. Can a comment be made in how these two relate evolutionarily? HTLV-1 is

closer to Lentiviruses and Beta-retroviruses when RT sequences are used to draw phylogenetic relationships. Is the same true for CA sequences?

Conclusions, line 242.

A logical inversion was made. The Gag lattice formation and maturation may impact transmission behaviour, not the other way around. The sentence may be rephrased as "The implication of the unique features of HTLV-1 Gag lattice stabilization showed here on HTLV-1 cell-cell transmission behaviour is unclear".

Figure S5B:

Please indicate in legend and figure if tubes are made of full-length CA or the truncated construct. Right now the figures suggests it is full-length, while the main text (line 129) suggests it is the truncated protein. Negative staining of CA tubes show remarkable flexibility, not usually seem in CA tubes of other retroviruses. Is this also a consequence of CA-CTD movement?

Reviewer #2:

Remarks to the Author:

In this paper, Obr et al describe cryoET and subtomogram averaging analysis of HTLV-I, a deltaretrovirus. Previous studies of representatives of the other retrovirus genera support the general model that the C-terminal domain (CTD) of the capsid (CA) protein is the major determinant of immature assembly. Here, the authors show that HTLV-I does not conform to this general model. CTD makes dimers but the dimers do not form long-range order. Rather, the N-terminal domain (NTD) of CA is the major determinant of immature assembly. The structural analysis clearly indicates that NTD makes hexamers connected into the immature lattice by strong trimer interface. Mutagenesis experiments (previously published, with some additional new data reported in this paper) are in good agreement and support the importance of the NTD.

The major significance of this paper is that it completes the list of structural analysis for all retroviral genera, and it shows HTLV-I is different from other retroviruses with regards to immature assembly. The paper starts strong, but fizzles in the end with lack of clarity and insufficient discussion of the details explaining why HTLV-I is different. As it stands, the authors simply demonstrate difference but not a clear explanation for why.

A number of additional issues need to be addressed before it can be published.

1) The significance of the nanoDSF experiments is unclear. Ideally, the samples should be analyzed at the same protein concentration (or the same tryptophan concentration) to facilitate comparison of the first derivative curves. Instead, the authors used 1 mg/mL, and so the CA, NTD and CTD proteins are at different molar concentrations. Are results the same when measurements are done at different protein concentrations? Also, the protein behaviors under different buffer conditions is not explained well. Why did the NTD alone show a drop in apparent T_m upon assembly? Shouldn't the assemblies have higher T_m than non-assembled proteins? What is the cumulant radius and what information does it provide on the samples? The authors claim that the aggregation/assembly propensity in full-length HTLV-I CA is "clearly derived" from the NTD and not the CTD – this is an overinterpretation unless they also perform comparative analysis of other retrovirus CA where the opposite is true.

2) Related to the above, what are the assemblies shown in Fig S5? Are these immature or mature? Since the paper is focused on the immature virus structures, this validation is required.

3) There is an imbalance in the amount of biological validation of importance of the NTD vs CTD. The authors should describe in more detail how their structural analysis relates to published mutagenesis studies of the CTD. Is the dimer important?

4) The authors should provide additional discussion on the CTD configurations in immature HTLV-I vs other retroviruses. In other retroviruses, the CTD makes an immature hexamer through interactions of MHR and 6HB. What is the explanation for the why this CTD does not make hexamers? Is it the lack of 6HB element? Are there features in the MHR that explain why this would not mediate hexamerization in this case?

5) Authors should discuss if this is generalizable across deltaretrovirus genus. What if HTLV-I is unique across all retroviridae? Are the NTD and CTD features they describe here conserved in other members of this genus?

More minor:

6) Figure 2 should contain a panel similar to Fig 1B, which provides a guide connecting the image to the radial density profile

7) Figure S3 also describes the major conclusion of the paper, and should be a main figure (perhaps as an additional panel in Fig 4)

Reviewer #3:

Remarks to the Author:

In this study, the authors perform cryoEM analysis of the immature Gag lattice of HTLV-1 virus-like particles (VLPs) produced both in cells and in vitro. In the case of previously studied retroviruses, the immature Gag lattice is stabilized primarily by interactions mediated by the capsid C-terminal domain (CA-CTD). In contrast, here the authors find that in the case of HTLV-1, Gag-Gag interaction in the immature lattice are stabilized primarily by the CA N-terminal domain (CA-NTD).

This work adds important new insights into the structural determinants of retroviral particle assembly, and highlight how different retroviruses use their structurally similar CA domains in strikingly different ways to build the immature Gag lattice that drives virus particle formation. The structural aspects of the work are straightforward and expertly performed, and the data are of high quality. The manuscript itself is clearly written and the work will be of significant interest to a broad audience. I have only relatively minor comments.

1. Fig. S8. The authors claim that mutations in the trimeric CA-NTD interface cause defects in virus particle assembly, implicating this interface in immature lattice formation. However, in several cases, the mutations appear to have a profound effect on Gag expression/stability rather than assembly per se. This needs to be clarified and discussed.
2. The variable distance between the CA layer and the viral membrane is very intriguing. Because the CA-NTD is connected to MA, this implies either that the MA layer undergoes significant structural change to accommodate highly variable MA-CA separation, or that in

regions in which the CA layer is far from the membrane the MA layer will locally detach from the membrane. Can the authors speculate on this point?

Author Rebuttal to Initial comments

We thank the reviewers for their evaluation of our manuscript, and their comments and suggestions.

We provide responses to each point below.

Reviewers' Comments:

Reviewer #1:

Remarks to the Author:

In this manuscript Obr and others present detailed and high-quality structural evidence on HTLV-1 immature lattice. These findings explain sui generis characteristics of HTLV-1 Gag protein organization, not found in other retroviruses. While these unique features were previously known through virology, fluorescence microscopy and cryoEM/ET data, there was no data at high-resolution available to mechanistically explain how these qualities were achieved until this manuscript. The data processing approach is on par with the most advanced data processing workflows of the field, which is corroborated by the resolution achieved in the presented subtomogram averaging maps. The quality of data presentation is good, but can be improved by the suggestions listed below in this review. The conclusions are appropriate and based on the structural findings produced by Obr and colleagues. Text is clearly written, and contextual information based on previous literature is well introduced. I recommend this manuscript to be accepted with minor revisions summarized below.

We thank the reviewer for these comments.

Figure 1:

Panel C shows the radial density plot of Gag lattice. The x axis shows distance in nm from the center of the box (I assume). To be more informative, the distance should state the distance from a known feature, such as the viral membrane. This would also facilitate comparison with similar radial density profiles from the literature. As it stands, the distance from the center of the box makes such interpretations and comparisons more difficult than it should be.

We realize that our description of panel C in Figure 1 was ambiguous. In the original figure we did not set the center of the box as the zero coordinate. Instead, we have reported the distance from local density minimum between the CA-NTD and CA-CTD, where the linker region between the CA-NTD and CA-CTD is located. We deemed this to be the best possible option as we report a similar radial density plot in Figure 2B for the MA₁₂₆CANC tubes, which do not have a membrane. Hence, centering the plot always between the two CA domains allowed the most straightforward comparison between the Gag VLPs and tubes.

In order to now allow a more straightforward comparison to similar profiles from literature, we also have included a second X-axis in Figure 1C indicating the distance with respect to the viral membrane. We have updated Figure 1 and its legend accordingly (**changes in bold**).

Figure 1: Cryo-ET of immature HTLV-1 Gag VLPs

A) Computational slice (8.8 nm thickness) through a cryo-electron tomogram containing HTLV-1 Gag-based VLPs. Protein density is black. Scale bar is 50 nm. B) Enlarged view of the Gag lattice within VLPs, as annotated by a white rectangle in panel (A). The arrowheads designate the different layers of the radially aligned Gag lattice underneath the viral membrane (VM). NC-RNP (1), CA-CTD (2), CA-NTD (3), MA (4), inner leaflet (5), outer leaflet (6). C) 1D radial density plot of the Gag lattice in immature HTLV-1 Gag-based VLPs. Two zero-value reference points are reported for the distance measurement. The primary X-axis reference 0 value is set at the viral membrane (top X-axis). The secondary X-axis reference 0 value is set at the local density minimum between the CA-NTD and CA-CTD (bottom X-axis) and is provided to allow a straightforward comparison to Figure 2C. The distance of the different layers from these reference points is given in nm. The annotation with arrowheads is as in panel (B). a.u. arbitrary units. D-F) Isosurface representations of the subtomogram average of the CA hexamer from HTLV-1 Gag-based VLPs. The CA-NTDs of the central hexamer are colored cyan, with one monomer highlighted in blue. Two additional CA-NTD monomers from adjacent hexamers are also colored in cyan, to highlight the trimeric inter-hexamer interface. The CA-CTDs of the central hexamer are colored orange/red. The hexameric arrangement is indicated by a small hexagon. D) CA lattice as seen from the outside of the VLP and rotated by 90 degrees to show a side view. E) Top view of the CA-NTD, with the trimeric inter-hexamer interface and the intra-hexameric interfaces indicated with a dashed red triangle and dashed yellow hexagon, respectively. CA monomers in the trimeric inter-hexamer interface are annotated with I, II and III. F) Top view of the CA-CTD hexamer. A dashed orange ellipsoid highlights one CA-CTD dimer linking adjacent hexamers. G-H) Molecular models of the CA-NTD and CA-CTD rigid body fitted into the EM-density of the immature CA lattice. Coloring as in D-F. G) Trimeric CA-NTD interactions linking hexamers, involving residues spanning helices 4 and 5. H) Interactions around the hexamer, involving helices 1,2 and 3 from adjacent CA-NTDs. I) Model of the CA-CTD dimer.

The same can be said from panel B in Figure 2 (in this case, the distance can be represented in relationship to the CA NTD layer/outer layer of tube).

Please see our answer to the comment above. We have kept Figure 2C as is to enable a straightforward comparison to Figure 1C.

Figure 1 legend:

In the description of E), replace “dashed yellow hexamer” for “dashed yellow hexagon”.

We have changed the description to hexagon.

Also remove the reference to “annotated as described in the main text”. There is no description in main text, instead, the description is on the legend of panels D-F.

We thank the reviewer for noticing this error. We have changed the figure legend accordingly to now properly explain the annotation in the figure legend.

Changes in bold: **“CA monomers in the trimeric inter-hexamer interface are annotated with I, II and III.”**

In description of G) add mention to the NTD helices involved in the trimeric interface (helices 4 and 5). This will also make the legend consistent to H) description.

We have done this as suggested.

Changes in bold: **“G) Trimeric CA-NTD interactions linking hexamers, involving residues spanning helices 4 and 5.”**

Figure 2:

Panel A should have a box showing general placement for the radial density plot, similar to the one showed in Figure 1A.

Panel B suffers from the same issue as described previously for Figure 1C. Distances in x axis should be represented in relationship to the CA NTD layer/outer layer of tube.

We have now added an additional panel to Figure 2 to show the placement of the radial density plot. We have updated the figure legend accordingly (see below, changes in bold). Concerning the comment on old Figure 2 panel B (the distance measurement in the radial density plot), we refer the reviewer to our reply to the comment on Figure 1.

Figure 2: Structural model of the immature HTLV-1 CA-NTD interactions

A) Computational slice (8.8 nm thickness) through a cryo-electron tomogram containing HTLV-1 MA₁₂₆CANC tubes. Protein density is black. Scale bar is 50 nm. **B)** Enlarged view of the CANC lattice within tubes, as annotated by a white rectangle in panel (A). The arrowheads designate the different layers of the tubular lattice NC-RNP (1), CA-CTD (2), CA-NTD (3). **C)** 1D radial density plot of the CANC lattice in MA₁₂₆CANC tubes, measuring the distance of the individual CA domains and NC-RNP from the linker between the CA-NTD and CA-CTD. **D)** EM density map of the immature HTLV-1 CA-NTD hexamer at 3.4 Å resolution, seen from the outside of the tube (left) and a side view (right). The three symmetry-independent trimer positions are colored in cyan, blue and pink. The remaining three CA-NTD domains of the central hexamer are colored in dark grey. Note the missing density for the CA-CTD in the side view on the right. **E)** Zoom-in view into the CA-NTD intra-hexamer interface (annotated with a yellow dashed hexagon in (C)). Labeling as in Figure 1. Assembly-relevant residues and the corresponding helices are annotated. **F)** Zoom-in view into the trimeric CA-NTD interface (annotated with a red triangle in (C)). Residues within the trimeric interface, which have been analysed via mutagenesis experiments are shown and colored according to the indicated coloring scheme. For simplicity one CA-NTD domain with the highlighted and colored residues is shown in side view on the right, to allow an easier appreciation of the residue location within the CA-NTD. The C2-hexamer center of the isosurface view in (C) is annotated by a schematic hexamer and the trimeric interfaces in (C) and (E) are annotated by black triangles.

Panel C, right volume is not informative. Please change cropping of the lattice to show a side view showing the coloured volumes on the left/top volume. As it is, the grey density obscures them and prevents making any relationship between the monomers in side view.

We have done this as suggested.

Panel D depicting NTD hexamer interactions should be shifted to show more context between I and I'. M17 in I seems to be reaching to I', but the way the box is placed prevents making any conclusions or analysis.

We have done this as suggested.

Discussion, line 193-196

Obr and colleagues introduce an interesting correlation between HTLV and MLV CA-NTD arrangements. Can a comment be made in how these two relate evolutionarily? HTLV-1 is closer to Lentiviruses and Beta-retroviruses when RT sequences are used to draw phylogenetic relationships. Is the same true for CA sequences?

The reviewer raises an interesting point here. Using gag and pol sequences for establishing evolutionary relationships, *Lentivirus* and *Betaretrovirus* genera are indeed closer to deltaretroviruses than gammaretroviruses (for example as reported in Llorens et al, 2008, DOI:10.1186/1471-2148-8-276, see attached figure 1 from their publication below).

Figure and figure legend from Llorens et al, 2008, DOI:10.1186/1471-2148-8-276

Figure legend quoted from Llorens et al: “**Phylogenetic analyses.** A) *Ty3/Gypsy* and *Retroviridae* phylogeny inferred based on the concatenated analysis of both gag and pol polyproteins. This tree is robust as gag and pol signals complement and correct each other. It also supports with significant bootstrap values the 2 groups of LTR retroelements and all their accepted lineages (clades, genera and classes). An extended version of this tree facilitating names, lineages, hosts, and Genbank accessions of all retroelement taxa used is provided as the Additional file 1 accompanying this paper (see the Section "Sequences and databases" in Methods). Decomposition of gag-pol tree and analysis of its two components separately, reveals similar phylogenetic signal but conflicting evolutionary perspectives. B) The phylogenetic signal of the pol polyprotein is robust and therefore responsible for the current known taxonomy and classification of *Ty3/Gypsy* and *Retroviridae* LTR retroelements into lineages. C) The gag signal supports the clades, genera and classes described in each group, but does not supports the 2 groups. Gag tree outlines an alternative scenario that may relate each *Retroviridae* class with one or more *Ty3/Gypsy* lineages.”

We now also performed multisequence alignment (MSA) for different retroviral CA sequences and generated a phylogenetic tree (using clustalOmega and then iTOL).

Indeed, the CA relationship for gamma- and deltaretroviruses is slightly closer to each other than to lentiviruses, which is the other way than using full-length Gag comparisons.

We interpret this CA phylogeny analysis to be in line with our observation that the quaternary immature CA arrangement in HTLV-1 and MLV are more alike.

We have added a new Extended Data Figure and also added the above finding to the discussion.

“When using gag and pol sequences for establishing evolutionary relationships, Lentivirus and Betaretrovirus genera are closer to Deltaretrovirus than Gammaretrovirus (39). We performed a multiple sequence alignment (MSA) for different retroviral CA sequences and generated a phylogenetic tree to evaluate how these evolutionary relationships change when we take into account only CA (40, 41) (Extended Figure 9). Indeed, we find the CA relationship for gamma- and deltaretroviruses to be slightly closer to each other than to lentiviruses, which is the other way compared to full-length Gag comparisons. We interpret this CA phylogeny analysis to be in line with our observation that the quaternary immature CA arrangement in HTLV-1 and MLV are more alike.

webserver implementation of iTOL (<https://itol.embl.de/>). The UniProt accession IDs and the CA residues are indicated in the FASTA header on the right side. The boxed numbers indicate the branch length.
EIAVY – Equine infectious anemia virus, FIVPE – Feline immunodeficiency virus, HV2 – HIV2, HV1 – HIV1, ALVA – Avian leukosis virus, RSVP – Rous sarcoma Virus, MMTV – Mouse mammary tumor virus, JSRV - Jaagsiekte sheep retrovirus, MPMV – Mason-Pfizer monkey virus, KORV – Koalaretrovirus, FLV – Feline leukemia virus, MLVMS – Murine leukemia virus, BLVAU – Bovine leukemia virus, HTLV1A – HTLV1, HTLV2, HTLV32 – HTLV3

Conclusions, line 242.

A logical inversion was made. The Gag lattice formation and maturation may impact transmission behaviour, not the other way around. The sentence may be rephrased as “The implication of the unique features of HTLV-1 Gag lattice stabilization showed here on HTLV-1 cell-cell transmission behaviour is unclear”.

We have rephrased the sentence as suggested.

Figure S5B:

Please indicate in legend and figure if tubes are made of full-length CA or the truncated construct. Right now the figures suggests it is full-length, while the main text (line 129) suggests it is the truncated protein. Negative staining of CA tubes show remarkable flexibility, not usually seem in CA tubes of other retroviruses. Is this also a consequence of CA-CTD movement?

The tubes shown in Figure S5B (new Extended Data 4) are assembled from full-length CA. We have now clarified this in the figure legend and also provide this info again in the methods.

“Cloning of truncated Gag constructs for in vitro assembly

The sequences encoding HTLV-1 MA₁₂₆CANC (Gag residues 126-344), HTLV-1 CA (131-344), HTLV-1 CA-NTD (131-258) and HTLV-1 CA-CTD (259-344) were cloned into pET28 expression vector in frame with 6xHis-SUMO tags using standard molecular cloning methods. The same procedure was done for cloning HIV-1 CA (133-363), HIV-1 CA-NTD (133-278) and HIV-1 CA-CTD (279-363).”

The observation of CA tube flexibility is interesting, but we acknowledge that the selected micrographs might have been misleading in this aspect. We have in the meantime repeated the HTLV-1 CA assembly several times, and flexible tubes are not a reproducible feature of this assembly. We can only speculate what might have caused the flexibility, but it likely could be derived from the negative staining procedure.

To avoid any confusion, we have replaced the micrographs in new Extended Data Figure 4B, now showing straight tubes.

Please also see our reply to reviewer 2, about whether these tubes are mature or immature assemblies.

Reviewer #2:

Remarks to the Author:

In this paper, Obr et al describe cryoET and subtomogram averaging analysis of HTLV-I, a

deltaretrovirus. Previous studies of representatives of the other retrovirus genera support the general model that the C-terminal domain (CTD) of the capsid (CA) protein is the major determinant of immature assembly. Here, the authors show that HTLV-I does not conform to this general model. CTD makes dimers but the dimers do not form long-range order. Rather, the N-terminal domain (NTD) of CA is the major determinant of immature assembly. The structural analysis clearly indicates that NTD makes hexamers connected into the immature lattice by strong trimer interface. Mutagenesis experiments (previously published, with some additional new data reported in this paper) are in good agreement and support the importance of the NTD.

The major significance of this paper is that it completes the list of structural analysis for all retroviral genera, and it shows HTLV-I is different from other retroviruses with regards to immature assembly. The paper starts strong, but fizzles in the end with lack of clarity and insufficient discussion of the details explaining why HTLV-I is different. As it stands, the authors simply demonstrate difference but not a clear explanation for why.

We thank the reviewer for the overall positive comments and helpful suggestions that have allowed us to further improve our manuscript.

Our paper provides the first structural explanation to many observations made by previous papers that reported the peculiar dependence of HTLV-1 assembly on the N-terminal CA domain (as also specifically noted by reviewer 1). Considering this we are confident that we do not simply demonstrate a difference of HTLV-1 to other retroviruses, but now provide a clearer view on how these previous observations have established HTLV-1 assembly as a particular case in the retrovirus family.

However, we agree with reviewer 2 that the implication of our observed unique features of HTLV-1 Gag lattice stabilization on the mechanism of virus spread remain unclear (please see also our reply to reviewer 1 and the logical inversion that was made in the initial manuscript). This still does not invalidate the importance of our findings but rather underlines the importance of future work on HTLV-1 transmission (e.g. the peculiar mode of cell-to-cell transfer) and to study HTLV-1 assembly for example in cells.

A number of additional issues need to be addressed before it can be published.

1) The significance of the nanoDSF experiments is unclear. Ideally, the samples should be analyzed at the same protein concentration (or the same tryptophan concentration) to facilitate comparison of the first derivative curves. Instead, the authors used 1 mg/mL, and so the CA, NTD and CTD proteins are at different molar concentrations. Are results the same when measurements are done at different protein concentrations? Also, the protein behaviors under different buffer conditions is not explained well. Why did the NTD alone show a drop in apparent T_m upon assembly? Shouldn't the assemblies have higher T_m than non-assembled proteins?

The reviewer raises several important points, which we try to address one by one below. As will be explained below, we have reorganized the entire section describing the NanoDSF experiments, due to additional experiments done with HIV-1 CA, CA-NTD and CA-CTD constructs for comparison. This resulted in moving the NanoDSF results to Extended Data Figure 8 and to also provide a Supplementary Table (S3) for easier interpretation.

In order to also simplify the representation of the NanoDSF data, we decided to remove the measurements for proteins in storage buffer, as we realized that they are not actively contributing to a better understanding of properties of CA and its domains. Accordingly, we have changed the naming of the two remaining tested conditions to "pre-assembly" and "post-assembly". A detailed explanation is given below.

Are results the same when measurements are done at different protein concentrations?

As suggested by the reviewer, we have performed the measurements for HTLV-1 CA, CA-NTD and CA-CTD at the same molar concentrations in the storage buffer and the pre-assembly state. The measurements for proteins after assembly were already done at different protein concentrations, as we here used the conditions, we knew would allow HTLV-1 CA tubes to form. We acknowledge that our description in the initial version of the manuscript has been insufficient. We updated the methods section accordingly.

The new measurements with the same molarity in the storage buffer and assembly buffer (prior to assembly) yielded almost identical results as for our initial measurements using 1mg/ml (see next page for reviewer figure R-1 comparing our initial data and the new data measured at the same molarity).

We also repeated the pre-assembly data measurements, normalizing the concentration on the tryptophan content in the respective constructs, with Tryptophan being the dominant UV-absorbing amino acid, for each of these constructs. Again, the results were similar to the initial measurements and the measurements at the same molarity.

Given these results, we decided to use NanoDSF data normalized for Tryptophan content when presenting the pre-assembly data in the manuscript. This is described in the methods section the following (changes in bold):

"Assembly reactions and preparation for nano differential scanning fluorimetry

*HTLV-1 and HIV-1 CA, CA-NTD, and CA-CTD were separately concentrated in Pierce concentrator tubes (10 MWCO for CA and CA-NTD, and 3 MWCO for CA-CTD). **Concentrations of monomeric HTLV-1 CA, CA-NTD, and CA-CTD were normalized to the tryptophan content of the respective constructs, with 0.96 mg/mL for CA, 1.15 for CA-NTD and 0.70 for CA-CTD under pre-assembly conditions (50 mM MES, 200 mM NaCl, pH 6) in a volume of 10 μ l and placed in Prometheus Series High Sensitivity Capillaries (NanoTemper, Cat. no. PR-C006). The same was done for HIV-1 CA (0.82 mg/mL), CA-NTD (0.66 mg/mL), and CA-CTD (2 mg/mL) but with a different pre-assembly buffer (50 mM MES, 150 mM NaCl, 5 mM β -mercaptoethanol, pH 6).***

*For measurement of HTLV-1 post-assemblies the proteins were first placed in assembly buffer (50 mM MES, 200 mM NaCl, pH6) **with a final concentration of ~680 μ M** and were incubated at 4°C for 4h and 26°C overnight for assembly to occur. A volume of 10 μ l was then placed in Prometheus Series High Sensitivity Capillaries (NanoTemper, Cat. no. PR-C006). For visualizing assemblies of HTLV-1 CA on a Tecnai T10 TEM an aliquot was used for negative staining with 2% Uranylacetate (UA).*

Similarly, for measurement of HIV-1 post-assemblies, protein was concentrated to 4 mg/mL (150 μ M) and dialyzed into 50 mM MES, 150 mM NaCl, 5 mM β -mercaptoethanol, pH 6, overnight. IP6 was then added to a final concentration of 4 mM and the protein was incubated at 30°C for 2h. A volume of 10 μ L was then placed in Prometheus Series High Sensitivity Capillaries (NanoTemper, Cat. no. PR-C006). For visualizing assemblies of HIV-1 CA on a Tecnai T10 TEM an aliquot was used for negative staining with 2% UA.”

Figure R-1: Comparison of HTLV-1 NanoDSF measurements done at 1mg/ml (left) and with the same molar concentration (right).

The measurements shown on the left are from the initially submitted version of the manuscript (old Figure 3). While the measured temperatures for T_m and inflection point are slightly lower in the new measurements using the same molarity, the overall curves and trends are highly similar between left and right experiments. Please also compare to new Extended Data Fig. 8, which shows the measurements from Tryptophan-normalized concentrations of proteins

Also, the protein behaviors under different buffer conditions is not explained well. Why did the NTD alone show a drop in apparent T_m upon assembly? Shouldn't the assemblies have higher T_m than non-assembled proteins?

This is indeed an interesting observation. Both full-length HTLV-1 CA and HTLV-1 CA-NTD show a drop in T_m upon assembly, which can seem counter-intuitive at first. While we do not have an unambiguous experimental explanation for this behavior, we believe their drop in T_m is due to the changed reactivity of CA and CA-NTD in assembly conditions. Specifically, the buffer conditions selected for assembly are chosen to stimulate protein-protein interactions (e.g. rendering the proteins to be more reactive towards assembly), which could make them both more dynamic and also flexible in order to allow forming such interactions. However, an assembly is not necessarily representing a stable end state (particularly for viruses, where CA interactions need to be metastable in order to allow eventual release of the viral genome). The observation of a drop in T_m might simply reflect the propensity of the proteins to be more reactive in both directions (forming metastable reactions that allow VLP assembly and disassembly).

Please also see our measurements for HIV-1 CA and CA-NTD, which show the same trend in T_m -drop post-assembly.

We now added this explanation to the revised manuscript in the Figure legend of Extended Data Figure 8.

“Both full-length HTLV-1 CA and HTLV-1 CA-NTD show a drop in T_m upon assembly. The observation of this drop in T_m might reflect the changed reactivity of CA and CA-NTD in assembly conditions. Specifically, the buffer conditions selected for assembly are chosen to stimulate protein-protein interactions (e.g. rendering the proteins to be more reactive towards assembly). This could make them both more dynamic and also flexible in order to allow forming such interactions. Virus assembly is not necessarily representing a stable end state, as viruses need to be metastable in order to allow eventual release of the viral genome. The observation of a drop in T_m might therefore reflect the propensity of the proteins to allow assembly and disassembly.”

What is the cumulant radius and what information does it provide on the samples?

The cumulant or hydrodynamic radius (r_H) shows the size of particles in solvated state. Assuming that CA and CA-NTD domain form oligomers upon assembly, an increased r_H reflects the propensity of proteins to form (specific or unspecific) interactions. We note that using this method one is not able to clearly define if one observes a true oligomeric assembly or aggregation, but given the shown behaviour of HTLV-1 CA to form tubes in the chosen conditions, we believe that assuming a somewhat specific protein-protein interaction leads to the increase in r_H .

Please see below also the exact definition of the cumulant radius as described in the manual of the NanoTemper Prometheus Panta system device:

“Cumulant fit definition:

The cumulants analysis was developed by Koppel in 1972 and is one of the most commonly used methods for DLS data analysis. It models the ACF using an average diffusion coefficient to obtain a single averaged Hydrodynamic radius r_H with a single Polydispersity index (PDI). “

The authors claim that the aggregation/assembly propensity in full-length HTLV-I CA is “clearly derived” from the NTD and not the CTD – this is an overinterpretation unless they also perform comparative analysis of other retrovirus CA where the opposite is true.

Our conclusions in the initial version of the manuscript were based on our own HTLV-1 NanoDSF measurements and by comparing it to the published NanoDSF results for HIV-1 (see PMID: 33115869).

As suggested by the reviewer, we have now performed additional experiments with HIV-1 CA, CA-NTD and CA-CTD to provide a comparative analysis. These results led us to change our conclusions of the nanoDSF experiments and we have changed the text and figures accordingly.

Specifically, the nanoDSF experiments for HIV-1 CA, HTLV-1 CA and their domains gave comparable results. This required us to change our line of argumentation about the aggregation and assembly propensity of HTLV-1 CA. However, the below described observations are still highly interesting as they argue for different mechanisms outside of the individual CA domains to regulate the assembly features in these different retrovirus species.

Changes in bold:

*“Given the peculiar behavior of the CA-CTD within the immature lattice, we sought to further characterize full-length CA and its two domains separately using biophysical approaches. To this end, HTLV-1 CA (residues 131-344), CA-NTD (residues 131-258) and CA-CTD (residues 259-344) (**Extended Data Fig. 4A**) were expressed in *E. coli* and purified. The CA, CA-NTD and CA-CTD proteins were subjected to differential scanning fluorimetry (nanoDSF), which monitors changes in intrinsic tryptophan fluorescence (ITF) upon protein unfolding as a function of temperature (38) (**Extended Data Fig. 8**, see also **highlighted tryptophan residues within HTLV-1 CA in Extended Data Fig. 1B**). We also conducted back-reflection and dynamic light scattering (DLS) experiments to measure turbidity and determine cumulant radius as a means of informing about aggregation and assembly properties. **We aimed to characterize the properties of CA and its domains in both a soluble form, as well as potentially assembled state. To this end, we carried out the measurements at two different time-points and protein concentrations: 1) a pre-assembly condition, directly after transferring the protein to assembly buffer at a concentration normalized for tryptophan content, and 2) a post-assembly condition, after incubation under assembly conditions. We note that we were able to obtain reproducible assemblies only for HTLV-1 CA (Extended Data Fig. 4B), but not its domains.***

*This data revealed overall similar behavior of full-length CA and CA-NTD, while CA-CTD behaved differently. CA-CTD reproducibly showed the highest melting temperature (T_m) in both conditions, suggesting that CA-CTD is the most stable of the three protein variants we tested (**Extended Data Fig. 8A, Supplementary Table S3**). Both full-length CA and CA-NTD showed a higher propensity to either aggregate or potentially form regular structures (see **Extended Data Fig. 8B-C**), while CA-CTD did not show a **substantial** increase in turbidity or cumulant radius in any of the conditions tested.*

For comparison of our HTLV-1 measurements to another retrovirus species, we repeated the same set of experiments with purified HIV-1 CA (residues 133-363), CA-NTD (133-278) and CA-CTD (279-363) (Extended Data 4C, Extended Data Fig 8D-F, Supplementary Table 3). Interestingly, HIV-1 CA and its domains behaved almost identically to their HTLV-1 counterparts in the different tested conditions, showing similar trends for melting curves as well as turbidity and cumulant radius. The minor differences observed were in slight changes in the respective inflection point temperatures and the signal strength for fluorescence ratio, most likely caused by the number of Tryptophan residues and their location within the individual CA constructs.

Again, as expected we were only able to reproducibly obtain assemblies, which could be visualized by negative staining TEM, for HIV-1 CA, but not the CA-NTD or CA-CTD alone (Extended Data Fig. 4D).”

Extended Data Fig. 8: Biophysical analysis of HTLV-1 and HIV-1 CA constructs

Biophysical analysis of HTLV-1 (panels A-C) and HIV-1 (panels D-F) CA and their N-terminal and C-terminal domains. Measurements are always performed in two conditions, termed pre-assembly (left side of respective panel) and post-assembly (right side of respective panel). See main text for further explanation on these conditions.

A,D) First derivative of the fluorescence ratio over a temperature ramp, measured by nanoDSF, for HTLV-1 (A) and HIV-1 (D) CA (green), CA-NTD (cyan), and CA-CTD (orange) in pre-assembly and post-assembly conditions. **B,E)** Turbidity change as a function of temperature, measured by back-reflection, for HTLV-1 (B) and HIV-1 (E) CA, CA-NTD, and CA-CTD in pre-assembly and post-assembly conditions. The melting temperatures (T_m) for all constructs as well as the turbidity inflection points are given in Supplementary Table S3.

C,F) Cumulant radius over a temperature ramp, measured by DLS, for HTLV-1 (C) and HIV-1 (F) CA, CA-NTD, and CA-CTD in pre-assembly and post-assembly conditions. CA and CA-NTD show an increase in cumulant radius coinciding with the observed turbidity increase and respective melting temperatures. CA-CTD shows no significant cumulant radius increase across all conditions. The color code is indicated on the bottom of the figure.

	HTLV-1			HIV-1		
	CA	CA-NTD	CA-CTD	CA	CA-NTD	CA-CTD
Pre-assembly T _m [°C]	51.83	55.61	57.96	47.25	49.86	63.62
Post-assembly T _m [°C]	47.09	48.95	57.25	44.63	46.36	62.66
Pre-assembly Turbidity IP [°C]	48.05	55.35	-	47.64	50.24	-
Post-assembly Turbidity IP [°C]	44.97	46.27	-	44.06	45.90	-

Table S3: Melting temperatures (T_m) and Turbidity inflection points (IPs) of HTLV-1 and HIV-1 CA constructs compared in pre-assembly and post-assembly conditions.

Taken together, these experiments indicate that the different assembly properties of HTLV-1 and HIV-1 are not necessarily derived solely by features within the individual CA-NTD domains, but must be influenced by a defined interplay between the CA domains, and their neighboring domains, such as NC and MA. Specifically, analyzing CA and its domains via NanoDSF does not necessarily consider immature or mature CA behaviour, but only provides insight into protein stability and its propensity to interact or aggregate. Hence, we conclude that the biophysical experiments provide interesting comparative insights into HTLV-1 and HIV-1 CA behavior, but do not necessarily allow further conclusions on how the proteins themselves define immature or mature assembly characteristics.

Given these additional results, we added a short section to the discussion with mentioning the above points.

“Physical properties of CA domains do not explain differences in assembly behavior between genera

Our biophysical experiments of HTLV-1 and HIV-1 CA and their respective domains indicate that the different assembly properties of HTLV-1 and HIV-1 are not derived solely from features within the individual CA-NTD domains, but must be also influenced by a defined interplay between the CA domains. In addition, it is likely that their context within Gag and the interaction with neighboring domains, such as NC and MA, plays an important role.

NanoDSF does not necessarily consider immature or mature CA behavior, but only provides insight into protein stability and its propensity to interact or aggregate. Hence, we conclude that the biophysics experiments done here provide interesting comparative insights into HTLV-1 and HIV-1 CA, but do not necessarily allow further conclusions on how the proteins themselves define immature or mature assembly characteristics.

Future experiments will be required to understand the differential contribution of CA domains to immature assembly in retroviruses. One potential avenue could be to determine high-resolution structures of HTLV-1/HIV-1 CA chimeras (42), and to solve the structure of the MA lattice in HTLV-1, similarly as done recently for HIV-1 (18).

.”

We thank the reviewer for raising this important point and suggesting this experiment, which made a substantial contribution to comparing CA behaviour by NanoDSF from different retrovirus species.

2) Related to the above, what are the assemblies shown in Fig S5? Are these immature or mature? Since the paper is focused on the immature virus structures, this validation is required.

The tubes shown in old Figure S5 (now Extended Data Fig. 4B) are immature HTLV-1 CA assemblies, based on a low-resolution subtomogram average of the CA layer of these tubes we have obtained. Specifically, the CA density, showing two distinct layers, resembles the immature architecture we observe in the MA₁₂₆CANC tubes.

Since the lattice arrangement was similar in case of both CA and CANC tubes, and since we already had solved the structure of CANC tubes to high-resolution, we did not follow up the structure from CA tubes. Noteworthy, solving a high-resolution structure of these HTLV-1 CA tubes is not straightforward and we hope the reviewer understands that this would be beyond the scope of this revision.

Hence, we provide the low-resolution subtomogram average in a figure for the reviewer (Figure R01), but refrain from including it in the manuscript. For comparison and to further underline the differences between mature and immature tube retrovirus lattice architecture, we are also showing subtomogram averages of *in vitro* assembled RSV (from Obr et al, 2021; doi.org/10.1038/s41467-021-23506-0) and EIAV CANC (from Dick et al, 2020; doi:10.1371/journal.ppat.1008277) constructs, forming mature- and immature-like morphology tubes, respectively.

Figure R01: Low-resolution structure of HTLV-1 CA tubes

A) Structure the immature HTLV-1 MA₁₂₆CANC lattice as determined in this study (see Figure 2). The lattice is shown from the outside of the tube (left) and as viewed down the tube axis (right). The two layers forming the CA-NTD (annotated in cyan) and CA-CTD (annotated in orange) are clearly visible.

B) Structure of the HTLV-1 CA lattice, from tubes as shown in Extended Data Figure ??, determined by subtomogram averaging. A similar lattice architecture as obtained for HTLV-1 MA₁₂₆CANC tubes is seen, indicating the HTLV-1 CA tubes to be immature.

C) For comparison the immature lattice structure of EIAV CANC tubes, as published by Dick et al, 2020 is shown)

D) For comparison a mature lattice structure of RSV CANC tubes is also shown, where the CA-NTD and CA-CTD layers are much closer to each other, typical for a mature assembly.

Please note that the structures in A-D are not shown to scale.

3) There is an imbalance in the amount of biological validation of importance of the NTD vs CTD. The authors should describe in more detail how their structural analysis relates to published mutagenesis studies of the CTD. Is the dimer important?

In our manuscript we have cited relevant mutagenesis data for the HTLV-1 CA-CTD, which is admittedly very scarce. Specifically, one publication that looked at HTLV-1 CA-CTD mutagenesis is a study by Rayne et al, 2001 (PMID: 11333909). In this paper, the mutagenesis experiments showed that the CA-NTD is important for Gag-Gag oligomerization, while the CA-CTD mutations had a substantially lesser effect. Importantly, we are not aware of literature that has mutated the CA-CTD dimer in HTLV-1.

Our story focused on CA-NTD, because 1) it showed an unprecedented feature, unseen in other retroviruses, and 2) there was already a wealth of mutagenesis experiments performed on CA-NTD. While the importance of CA-CTD is a relevant question, given the scarcity of mutagenesis data published so far, addressing this question would mean changing the scope of our manuscript in a major way.

We therefore find it preferable, if the role of CA-CTD dimer in HTLV-1 is addressed comprehensively in a separate manuscript.

We now explicitly state in the discussion that additional mutagenesis studies focusing on HTLV-1 CA-CTD would be an interesting avenue for future research.

“Additional studies focusing on the HTLV-1 CA-CTD, in particular considering the lack of exhaustive CA-CTD mutagenesis experiments, are warranted to more accurately determine the interactions in this region of the immature lattice. “

4) The authors should provide additional discussion on the CTD configurations in immature HTLV-I vs other retroviruses. In other retroviruses, the CTD makes an immature hexamer through interactions of MHR and 6HB. What is the explanation for the why this CTD does not make hexamers? Is it the lack of 6HB element? Are there features in the MHR that explain why this would not mediate hexamerization in this case?

In the initially submitted version of our manuscript we have already tried to provide such an explanation. Specifically, we aimed to elaborate on the fact the residues in the MHR are not positioned to form interactions around the hexameric ring. We also stated that there is no 6HB element visible in our HTLV structures. Please, see also Extended Data Fig. 3, where all determined immature retrovirus structures, except HTLV-1, either have a 6HB or a density for a spacer peptide-like region.

When looking at the HTLV-1 CA-CTD model and also when comparing the MHR sequence of different retrovirus species using MSA (please see also our reply to reviewer 1), we cannot spot an immediate difference in the MHR sequence or structure (or the CA-CTD in general) that would allow us to conclude what might cause the HTLV-1 CA-CTD to not form hexamers. To further substantiate this observation, we provide Figure R02 for the reviewer that shows the MSA alignment result focused on the MHR of the studied viruses.

When investigating the MHR residues that could be positioned to establish interactions around the hexameric ring (annotated by arrowheads in the figure), we find that there is no clear pattern that would argue for any of these HTLV-1 residues to interfere with such an interaction.

Hence, considering the above information, we believe our explanations in the manuscript are already sufficient given that further speculations would require substantial mutagenesis analysis of the CA-CTD, which together with the analysis of the CA-CTD dimer makes a case for a separate manuscript.

Figure R02: MSA alignment of the MHR

MSA of the MHR in CA of 16 different retrovirus species, from different genera (see also Extended Data Fig. 9). The MSA was generated using ClustalOmega and displayed using JalView.

Residues in the MHR positioned to potentially establish interactions around the hexameric ring (using the immature HIV-1 CA-SP1 hexamer, pdb 5L93 as orientation reference) are annotated with black arrowheads.

5) Authors should discuss if this is generalizable across deltaretrovirus genus. What if HTLV-I is unique across all retroviridae? Are the NTD and CTD features they describe here conserved in other members of this genus?

We agree with the reviewer, that it would be exciting to have the structure of another immature deltaretrovirus assembly. Indeed, this is part of a project we are actively working on. However, obtaining structures such as the ones presented in this manuscript is a non-trivial task, potentially taking several years (as was the case for HTLV-1, requiring establishing the *in vitro* assembly, data acquisition and image processing workflows). We hope that the reviewer understands that it would be beyond the scope of this manuscript to solve another deltaretrovirus structure.

In the conclusions section we mention that HTLV-1 might be unique and that looking at other deltaretroviruses would be highly interesting. We have rephrased this section slightly to state more explicitly that additional structures from immature deltaretrovirus assemblies would be helpful.

*“In addition to HTLV-1, the Deltaretrovirus genus contains multiple members, such as bovine leukemia virus. **Structural comparison of the immature Gag lattice in other deltaretroviruses** could help us better understand the general assembly principles within this genus, and could clarify whether the observed dominance of the HTLV-1 CA-NTD in the immature Gag lattice is conserved within the genus”.*

More minor:

6) Figure 2 should contain a panel similar to Fig 1B, which provides a guide connecting the image to the radial density profile

As also suggested by reviewer 1, we have included the additional panel to Figure 2. Please see the updated figure in the response to the comment from reviewer 1.

7) Figure S3 also describes the major conclusion of the paper, and should be a main figure (perhaps as an additional panel in Fig 4)

In line with this suggestion, we have reorganized the presentation of the data. Figure S3 is now main figure 3. In exchange, we have moved the NanoDSF characterization to be Extended Data Figure 8.

Reviewer #3:

Remarks to the Author:

In this study, the authors perform cryoEM analysis of the immature Gag lattice of HTLV-1 virus-like particles (VLPs) produced both in cells and in vitro. In the case of previously studied retroviruses, the immature Gag lattice is stabilized primarily by interactions mediated by the capsid C-terminal domain (CA-CTD). In contrast, here the authors find that in the case of HTLV-1, Gag-Gag interaction in the immature lattice are stabilized primarily by the CA N-terminal domain (CA-NTD).

This work adds important new insights into the structural determinants of retroviral particle assembly, and highlight how different retroviruses use their structurally similar CA domains in strikingly different ways to build the immature Gag lattice that drives virus particle formation. The structural aspects of the work are straightforward and expertly performed, and the data are of high quality. The manuscript itself is clearly written and the work will be of significant interest to a broad audience. I have only relatively minor comments.

We thank the reviewer for the encouraging comments.

1. Fig. S8. The authors claim that mutations in the trimeric CA-NTD interface cause defects in virus particle assembly, implicating this interface in immature lattice formation. However, in several cases, the mutations appear to have a profound effect on Gag expression/stability rather than assembly per se. This needs to be clarified and discussed.

We acknowledge that our description of which residues cause particle assembly defects has been not clear enough. We have now extended this section as stated below (Changes in bold):

“Double-alanine swap experiments on a set of adjacent residues at the trimeric CA-NTD interface were performed to characterize this interface. Residue pairs H72/Q73, Y90/N91, P92/L93, G95/P96, L97/R98, V99/Q100 were mutated (Figure 2E) and screened for altered particle production efficiency. **Previously, the G95/P96 pair was already shown in an insertion/substitution mutation approach to reduce particle formation (28), supporting the importance of this region in VLP assembly. Our mutagenesis experiments revealed a substantial role for the H72/Q73, P92/L93 and V99/Q100 residue pairs in immature Gag assembly and particle release (Extended Data Fig.7C). The Y90/N91, G95/P96, L97/R98 residue pair substitutions also reduced the particle production, but also significantly impacted cellular Gag levels, suggesting these mutations to also have an effect on expression or stability of the Gag protein. Overall, we conclude from these observations that residues within the trimer interface have important roles for both the behavior of the individual Gag molecules, as well as Gag lattice assembly and stability.**”

2. The variable distance between the CA layer and the viral membrane is very intriguing. Because the CA-NTD is connected to MA, this implies either that the MA layer undergoes significant structural change to accommodate highly variable MA-CA separation, or that in regions in which the CA layer is far from the membrane the MA layer will locally detach from the membrane. Can the authors speculate on this point?

The variable distance of the CA layer to the membrane is indeed intriguing. As correctly suggested by the reviewer, either MA detachment or a significant structural change of MA could accommodate this variability.

We currently do not have any data supporting MA detachment from the membrane in areas of increased flexibility. We regularly can see protein density attached to the inner leaflet of the viral membrane in areas where there is Gag (please also see Movie S1 and the deposited tomogram of HTLV-1 Gag VLPs, accessible via <https://www.ebi.ac.uk/emdb/EMD-17941>).

In addition, previous studies reported HTLV-1 MA to bind to membranes with higher affinity, and efficiency, but with lower specificity than other retrovirus MA (see PMID: 34298060; PMID: 21289126; PMID: 25491356).

This argues for MA to be able to potentially extend by undergoing a conformational change. However, we do not yet have structural or other data that would allow us to describe this in further detail. We hope to be able to solve the immature MA lattice of HTLV-1 in future projects in order to provide this kind of information, but this is beyond the scope of the current manuscript.

We have added a short section to the discussion

“The variable distance of CA to the membrane could be explained by either MA detachment or a significant structural change of MA. Our data do not provide any evidence for MA detachment from the membrane in areas of increased flexibility, as we regularly can see protein density attached to the inner leaflet of Gag-associated viral membrane (see Movie S1). In addition, previous studies reported HTLV-1 MA to bind to membranes with higher affinity, and efficiency, but with lower specificity (44–46), arguing against MA detachment from the viral membrane.”

Decision Letter, first revision:

Message: Our ref: NSMB-A47998A

29th Apr 2024

Dear Dr. Schur,

Thank you for submitting your revised manuscript "Unconventional stabilization of the human T-cell leukemia virus type 1 immature Gag lattice" (NSMB-A47998A). It has now been seen by the original referees and their comments are below. The reviewers find that the paper has improved in revision, and therefore we'll be happy in principle to publish it in Nature Structural & Molecular Biology, pending minor revisions to satisfy the referees' final requests and to comply with our editorial and formatting guidelines.

We are now performing detailed checks on your paper and will send you a checklist detailing our editorial and formatting requirements in about 2-3 weeks. Please do not upload the final materials and make any revisions until you receive this additional information from us.

Sincerely,

Katarzyna Ciazynska, PhD
(she/her)
Associate Editor
Nature Structural & Molecular Biology
<https://orcid.org/0000-0002-9899-2428>

Reviewer #1 (Remarks to the Author):

The authors have satisfactorily addressed the points raised in the review. I recommend the revised manuscript to be accepted.

Reviewer #3 (Remarks to the Author):

the authors have addressed my concerns

Final Decision Letter:

Message: 14th Aug 2024

Dear Dr. Schur,

We are now happy to accept your revised paper "Distinct stabilization of the human T-cell leukemia virus type 1 immature Gag lattice" for publication as an Article in Nature Structural & Molecular Biology.

Note the policy of the journal on data deposition:

<http://www.nature.com/authors/policies/availability.html>.

Your paper will be published online soon after we receive proof corrections and will appear in print in the next available issue. You can find out your date of online publication by contacting the production team shortly after sending your proof corrections.

You may wish to make your media relations office aware of your accepted publication, in case they consider it appropriate to organize some internal or external publicity. Once your paper has been scheduled you will receive an email confirming the publication details. This is normally 3-4 working days in advance of publication. If you need additional notice of the date and time of publication, please let the production team know when you receive the proof of your article to ensure there is sufficient time to coordinate. Further information on our embargo policies can be found here:

<https://www.nature.com/authors/policies/embargo.html>

Please note that *Nature Structural & Molecular Biology* is a Transformative Journal (TJ). Authors may publish their research with us through the traditional subscription access route

or make their paper immediately open access through payment of an article-processing charge (APC). Authors will not be required to make a final decision about access to their article until it has been accepted. Find out more about Transformative Journals

Sincerely,

Katarzyna Ciazynska, PhD
(she/her)
Associate Editor
Nature Structural & Molecular Biology
<https://orcid.org/0000-0002-9899-2428>